# FROM SELF-INCONSISTENCY TO STABILITY: ACHIEVING ORDER INVARIANT IN-CONTEXT LEARNING

## ABSTRACT

Large Language Models (LLMs) exhibit powerful reasoning capabilities, particularly when guided by in-context learning (ICL). However, their performance is brittle to demonstration order: accuracy can swing from perfect to random based solely on the permutation of input ordering. This sensitivity reveals a fundamental vulnerability where models rely on spurious positional correlations (noise) rather than semantic content (signal). To address this reliability gap, we introduce **Self-Inconsistency Optimization (SIO)**, a simple model-agnostic post-training framework that teaches models to focus on *what* is said, not *how* it is arranged. SIO generates semantically equivalent inputs through permutation and explicitly trains the model to align its output distributions using our proposed self-inconsistency loss which is based on the Jensen–Shannon divergence. We provide a theoretical justification for our framework, proving that minimizing this self-inconsistency loss is sufficient to achieve the desired order invariance. Furthermore, the Bayesian update design of SIO provides a stable optimization process by decoupling the model's prior knowledge from the alignment objective, allowing it to integrate seamlessly with existing post-training pipelines such as reinforcement learning. Empirical evaluations on mathematical reasoning benchmarks show that SIO substantially mitigates order sensitivity while maintaining or even improving task accuracy. Our source code is available at `https://anonymous.4open.science/r/From-Self-Inconsistency-to-Stability-E0BC`.

## 1 INTRODUCTION

Large Language Models (LLMs) have demonstrated remarkable capabilities, largely driven by in-context learning (ICL), which allows adaptation to new tasks with a few demonstration examples (Brown et al., 2020). This eliminates the need for costly retraining (Winata et al., 2023; King & Flanigan, 2023). However, ICL is notoriously sensitive to the order of these examples, a phenomenon we refer to as *positional noise*. As illustrated in Figure 1, an LLM's performance can vary dramatically with simple permutations of the demonstrations (Lu et al., 2022). This brittleness is a critical reliability issue, indicating that models learn from spurious positional noise rather than the intended semantics (Harutyunyan et al., 2024). Such unreliability is a significant barrier to deploying LLMs in high-stakes applications where consistent and trustworthy behavior is essential (Gero et al., 2023; Novikova et al., 2025; Bajwa et al., 2021).

This reliance on positional noise over semantic content is well evaluated and documented: Min et al. (2022) showed that LLM predictions are more sensitive to prompt formatting than to the correctness of the ground-truth labels in demonstrations, suggesting that models often prioritize superficial patterns over underlying meaning. Current methods to mitigate this issue have significant limitations. Inference-time strategies such as prompt optimization and decoding heuristics can improve performance over random orderings but do not address the model's fundamental reliance on positional noise (Lu et al., 2022; Wang et al., 2023; He et al.; Zhao et al., 2021; Zhang et al., 2024). On the other hand, existing training-based methods often require sophisticated designs such as distillation and adversarial training, which incur significant computational overhead; they are typically evaluated only on simple multiple-choice classification tasks and rarely offer formal guarantees of invariance beyond empirical agreement (Chen et al., 2025; Xiang et al., 2024; Liusie et al., 2024). Thus, this raises a natural question: ***How can we efficiently and theoretically ground a post-training approach to make LLMs robust to positional noise?***

Figure 1: **Our proposed framework makes LLMs robust to in-context example order.** The top panel shows that a standard pretrained LLM is sensitive to the order of demonstrations (D). Changing the order flips the model's prediction for the same question (Q) from correct (9) to incorrect (8). In contrast, the bottom panel shows that our Self-Inconsistency Optimization framework makes the LLM robust, consistently producing the correct answer regardless of the demonstration order.

To overcome these challenges, we introduce **Self-Inconsistency Optimization (SIO)**, a simple and theoretically grounded post-training framework designed to instill robustness. The principle is intuitive: semantically equivalent inputs with ICL samples, regardless of their example ordering, should yield consistent output distributions (Liusie et al., 2024). To enforce this consistency, we train the model to align its own output distributions across these equivalent inputs by minimizing a Jensen–Shannon Divergence-based loss. To achieve this, we introduce a novel Bayesian update training architecture that decouples the model's pretrained knowledge from the alignment objective of Group Sequence Policy Optimization (GSPO) loss (Korbak et al., 2022; Zheng et al., 2025). This separation is crucial for preserving task accuracy and resolving common training instabilities that arise from the conflicting objectives inherent in reinforcement learning losses (Shao et al., 2024). Our contributions are summarized as follows:

- **Self-Inconsistency Optimization.** We introduce SIO, a general model-agnostic framework that enables LLMs to process input robustly with reduced sensitivity to the order of in-context examples.

- **Theoretical Grounding for Invariance.** We provide a formal justification for our approach and demonstrate that its benefits extend beyond positional noise to other forms of noise, such as tokenization noise, thereby fundamentally improving model reliability.

- **Stabilized Training for Robust Performance.** We introduce a stable training architecture inspired by Bayesian updates that decouples the base model's knowledge from the alignment objective, resolving common post-training instabilities.

- **Empirical Evaluation.** We demonstrate the effectiveness of our framework in two state-of-the-art LLMs across three mathematical reasoning benchmarks, showing a consistent reduction in order sensitivity without degrading task performance.

## 2 PRELIMINARIES

This section establishes the formal notation used throughout the paper and defines the metrics for evaluating both task performance and invariance to the order of in-context demonstrations in LLMs.

### 2.1 NOTATIONS AND PROBLEM DEFINITION

We study *few-shot ICL under order invariance of demonstrations* caused by positional noise. Let $\pi_\theta$ denote an LLM policy parameterized by $\theta$, which induces a conditional distribution $P_\theta(\cdot \mid x)$ over response sequences given an input prompt $x$. An input prompt consists of three components: the task instruction $t$ describing the high-level objective, the query $q$ with corresponding ground-truth label $y$, and a set of demonstrations $D = \{d_1, \ldots, d_k\}$, where each example is a pair $d_j = (q_j, y_j)$. We define the *signal* required to solve the task as $S = (t, q, D)$, which is invariant to the ordering

of elements in $D$. In practice, however, prompts present demonstrations as a sequence, and each permutation induces a distinct *noisy ordering*. Let $D_i$ denote the $i$-th ordering of $D$, corresponding to *positional noise* $n_i$, and define the *permuted input* as $x_i = (t, D_i, q)$. For a given $x_i$, the model generates a set of candidate responses $\hat{Y}_i = \{\hat{y}_i^1, \ldots, \hat{y}_i^m\}$ with $\hat{y}_i^j \sim P_\theta(\cdot \mid x_i)$, where $\hat{Y}_i$ may vary across permutations despite encoding the same underlying signal $S$. The central challenge is to design methods that ensure the learned policy $\pi_\theta$ is *robust to positional noise*, so that predictions depend only on the semantic content of demonstrations rather than their ordering.

## 2.2 EVALUATION METRICS

Our primary goal is to achieve invariance to the order of demonstrations in ICL without compromising task accuracy. To comprehensively assess this, we measure not only task accuracy but also three distinct aspects of stability: correctness fluctuation (Pairwise Accuracy Difference), output set similarity (Jaccard Distance), and full output distributional divergence (Self-Inconsistency).

**Accuracy ($\uparrow$).** This metric evaluates the model's core task-solving capability using the **pass@m** success rate (Chen et al., 2021). A task is considered successfully solved if at least one of the $m$ generated responses for a given query matches the ground truth. The final accuracy is the average success rate across the entire test dataset $\mathcal{D}$. Formally, with $\mathbb{I}(\cdot)$ as the indicator function,

$$\text{Accuracy} = \mathbb{E}_{(q,y)\sim\mathcal{D}} \left[ \max_{j=1,\ldots,m} \mathbb{I}(\hat{y}^j = y) \right]$$

**Pairwise Accuracy Difference ($\downarrow$).** This metric measures the instability in correctness when the demonstration order is altered (Chen et al., 2022). For each query, we generate responses from two distinct permutations of the demonstrations, $x_1$ and $x_2$. We then compute the absolute difference between their pass@$m$ scores. A lower value indicates greater stability, with a score of 0 signifying that permuting the demonstrations does not change a correct answer to an incorrect one, or vice versa.

$$\Delta\text{Acc} = \mathbb{E}_{(q,y)\sim\mathcal{D}} \left[ \left| \max_j \mathbb{I}(\hat{y}_1^j = y) - \max_j \mathbb{I}(\hat{y}_2^j = y) \right| \right]$$

**Response Set Jaccard Distance ($\downarrow$).** This metric assesses the dissimilarity between the sets of generated answers resulting from different demonstration orders. For two distinct permutations, $x_1$ and $x_2$, we compute the Jaccard distance between their corresponding sets of unique generated responses, $\hat{Y}_1$ and $\hat{Y}_2$. The Jaccard distance is defined as 1 minus the Intersection over Union (IoU) (Jaccard, 1901).

$$\text{Jaccard Distance}(\hat{Y}_1, \hat{Y}_2) = 1 - \mathbb{E}_{(q,y)\sim\mathcal{D}} \left[ \frac{|\hat{Y}_1 \cap \hat{Y}_2|}{|\hat{Y}_1 \cup \hat{Y}_2|} \right]$$

A score of 0 indicates perfect invariance, meaning both permutations produce identical sets of answers. Conversely, a score of 1 implies the answer sets are completely disjoint.

**Self-Inconsistency ($\downarrow$)** To obtain a more holistic measure of stability, we introduce **Self-Inconsistency**, which quantifies the divergence in a model's behavior across multiple demonstration permutations. A truly invariant model should produce not only the same final answer but also a consistent reasoning process (e.g., Chain-of-Thought). Therefore, this metric compares the full output probability distributions, $P_\theta(\cdot|x_i)$, rather than just the final outputs.

We use the Generalized Jensen–Shannon Divergence (JSD) to measure the divergence among a set of probability (Englesson & Azizpour, 2021).

**Definition 1** (Generalized Jensen–Shannon Divergence). *Given a set of $N$ probability distributions $\{P_j\}_{j=1}^N$ and a corresponding weight vector $w = (w_1, \ldots, w_N)$ where $w_j > 0$ and $\sum_{j=1}^N w_j = 1$, the mixture distribution is defined as $M = \sum_{j=1}^N w_j P_j$. The Generalized JSD is the weighted sum of the Kullback-Leibler divergences ($D_{\text{KL}}$) from each distribution $P_j$ to the mixture $M$:*

$$\text{JSD}_w(P_1, \ldots, P_N) = \sum_{j=1}^N w_j D_{\text{KL}}(P_j \| M).$$

We define Self-Inconsistency as the JSD of the output distributions from a set of $N$ permuted inputs $\{x_1, \ldots, x_N\}$:

$$\text{Self-Inconsistency} = \text{JSD}_w\big(P_\theta(\cdot \mid x_1), \ldots, P_\theta(\cdot \mid x_N)\big)$$

By default, we use uniform weights ($w_j = 1/N$), treating each permutation as equally important. A lower Self-Inconsistency score signifies greater stability in the model's generative process.

# 3 SIO: SELF-INCONSISTENCY OPTIMIZATION

We introduce a framework that enforces order invariance in LLMs by finetuning the model so that its output distribution is conditionally independent of the ordering of in-context demonstrations. We begin by outlining the overall methodology and then provide the theoretical foundations.

## 3.1 OVERVIEW OF SIO

Our proposed methodology, illustrated in Figure 2, consists of three key components. First, we apply a permutation-based data augmentation strategy, generating multiple training samples from each data point by reordering its in-context demonstrations and producing corresponding model responses. Second, we adopt a dual-objective finetuning scheme with a tailored loss function that jointly optimizes task performance via reinforcement learning and enforces order invariance by minimizing the divergence between output distributions across permutations. Finally, we introduce a Bayesian update–based finetuning architecture that stabilizes training by decoupling the model's pretrained knowledge from the new alignment objectives.

## 3.2 DATA AUGMENTATION & GENERATION

To mitigate sensitivity to the ordering of demonstration examples, we reformulate each data point $(q, y)$ into a structured training instance $(X, Y, R)$, where $X$ denotes a set of augmented inputs, $Y$ the corresponding sampled responses, and $R$ the associated rewards. The training instances are constructed as follows. First, we create the augmented inputs $X = \{x_1, \ldots, x_N\}$ by generating $N$ unique permutations of the $k$ available demonstrations (where $2 \leq N \leq k!$) and concatenating each with the original instruction and query. Next, for each input $x_i \in X$, we sample $m$ responses from the model to form a response set $\hat{Y}_i = \{\hat{y}_i^1, \ldots, \hat{y}_i^m\}$. The union of these individual sets constitutes the overall response set, $Y = \{\hat{Y}_1, \ldots, \hat{Y}_N\}$. Finally, we evaluate each response $\hat{y} \in Y$ using a reward function (e.g., a binary score for correctness based on the ground truth $y$) to produce a corresponding reward, forming the final reward set $R$. This reward signal is crucial for preserving task performance during post-training.

## 3.3 TRAINING OBJECTIVES

With the augmented training data $(X, Y, R)$, we then finetune the model using two complementary objectives to mitigate the positional noise. The multiple objectives are shown as follows:

**Group Sequence Policy Optimization (GSPO).** To preserve task performance during post-training, we apply reinforcement learning using GSPO (Zheng et al., 2025), which aligns sequence-level probabilities with sequence-level rewards. For each augmented input set $X$, we have a group of $G = m \times N$ responses in $Y$. We define the sequence-level importance ratio as $s_i(\theta) = \frac{\pi_\theta(y_i|x)}{\pi_{\theta_{\text{old}}}(y_i|x)}$ and the group-standardized advantage as $\hat{A}_i = R_i - \bar{R}$, where $\bar{R}$ is the average reward of the group. The objective maximizes the clipped surrogate:

$$\max_\theta \mathcal{J}_{\text{GSPO}}(\theta) = \mathbb{E}_{x\sim\mathcal{D},\, \mathcal{G}\sim\pi_{\theta_{\text{old}}}(\cdot|x)}\left[\frac{1}{G}\sum_{i=1}^{G} \min\Big(s_i(\theta)\,\hat{A}_i,\ \text{clip}\big(s_i(\theta),\, 1-\epsilon,\, 1+\epsilon\big)\hat{A}_i\Big)\right].$$

Intuitively, this objective up-weights responses that outperform the group average ($\hat{A}_i > 0$) and down-weights those that underperform ($\hat{A}_i < 0$), while clipping stabilizes the training with controlled update. For minimization during training, we use the negative of the objective function, defining the loss as:

$$\min_\theta \mathcal{L}_{\text{GSPO}}(\theta) = -\mathcal{J}_{\text{GSPO}}(\theta)$$

Figure 2: The SIO pipeline. **Top (Data Generation):** Semantically equivalent inputs ($X$) are generated via permutation and passed to an LLM to create responses ($Y$), which are then scored by a reward function ($R$). **Bottom (LLM Training):** A pretrained LLM is frozen and augmented with trainable LoRA modules. The unembedding layer is split into a frozen *policy network* (prior) and a trainable *value network* (likelihood); their logits are summed for the final distribution $\pi_c$. The model is trained with a composite loss, including: (1) GSPO for reward alignment, (2) Distributional- and Self-Inconsistency losses for noise invariance, and (3) an L2 penalty for regularization.

**Distributional Inconsistency (DI) Loss.** To directly enforce invariance, we introduce a loss that minimizes the divergence between the output distributions of augmented inputs and a target mixture distribution formed from a frozen **reference policy** $\pi_{\mathrm{ref}}$ (the original pretrained model).

$$\min_{\theta} \mathcal{L}_{\mathrm{DI}}\big(\pi_\theta \,\|\, \pi_{\mathrm{ref}}\big) = \sum_{j=1}^{N} w_j \; \mathrm{D}_{\mathrm{KL}}\Big(P_{\pi_\theta}(\cdot \mid x_j) \,\Big\|\, M_{\mathrm{ref}}\Big), \quad \text{where } M_{\mathrm{ref}} = \sum_{j=1}^{N} w_j \, P_{\pi_{\mathrm{ref}}}(\cdot \mid x_j).$$

The weights $w_j$ can be uniform ($w_j = 1/N$) or set dynamically based on rewards. This objective encourages the model's output distributions for all permutations to converge towards a common, stable target, preventing catastrophic forgetting by anchoring the model to its original knowledge.

**Self-Inconsistency (SI) Loss.** We also penalize the residual divergence among the *online* distributions themselves to directly enforce consistency. This is a special case of the DI loss, where $\mathcal{L}_{\mathrm{SI}}(\pi_\theta) = \mathcal{L}_{\mathrm{DI}}\big(\pi_\theta \,\|\, \pi_\theta\big)$ with uniform weights ($w_j = 1/N$).

$$\min_{\theta} \mathcal{L}_{\mathrm{SI}}(\pi_\theta) = \mathrm{JSD}_w\big(P_{\pi_\theta}(\cdot \mid x_1), \ldots, P_{\pi_\theta}(\cdot \mid x_N)\big).$$

### 3.4 STABLE FINETUNING VIA BAYESIAN UPDATE

Standard reinforcement learning can exhibit instability due to a direct optimization conflict between improving on the reward and staying close to the original policy simultaneously by minimizing the surrogate loss and KL-divergence penalty against a frozen pretrained reference model on the same model output. To resolve this, we introduce a finetuning architecture grounded in Bayesian updating, which reframes post-training task such as alignment as $P(\text{posterior}) \propto P(\text{likelihood}) \times P(\text{prior})$. We implement this by conceptually splitting the model's final unembedding layer into two components. The first, representing the *prior*, is the frozen pretrained unembedding layer, which encapsulates the model's prior knowledge, $P(\text{response})$, and produces output logits $z_p$. The second component, representing the *likelihood*, is a new, trainable low-rank (LoRA) adapter that models the alignment preference, $P(\text{reward}|\text{response})$, and outputs logits $z_v$. The finetuned model's response is sampled from the posterior distribution, $P(\text{response}|\text{reward})$, which incorporates reward information. The final logits are computed as the sum $z_c = z_p + z_v$, which in log-space is equivalent to multiplying their respective probability distributions, directly implementing the Bayesian update (see Appendix B).

During training, only the Likelihood Network is updated. The posterior can be viewed as a Product of Experts (PoE) (Hinton, 2002), where a valid response must have high probability under both the prior (preserving foundational knowledge) and the likelihood (aligning with new preferences). This decoupling of objectives intrinsically mitigates model collapse and promotes stable training.

**Final Objective for SIO Training.** Let $\pi_p$ and $\pi_c$ be the probability distributions corresponding to the softmax of logits $z_p$ and $z_c$, respectively. Our final training loss combines all components:

$$\mathcal{L}_{\text{SIO}}(\theta) = \lambda_1 \, \mathcal{L}_{\text{GSPO}}(\pi_c) + \lambda_2 \, \mathcal{L}_{\text{DI}}(\pi_p \| \pi_{\text{ref}}) + \lambda_3 \, \mathcal{L}_{\text{SI}}(\pi_c) + \lambda_4 \, \|z_v\|_2^2,$$

where $\lambda_i$ are non-negative hyperparameters. $\mathcal{L}_{\text{GSPO}}$ learns the task preference, $\mathcal{L}_{\text{DI}}$ regularizes the prior network to be self consistent while retaining the pretrained knowledge, $\mathcal{L}_{\text{SI}}$ enforces invariance on the final output, and an L2 penalty regularizes the likelihood network. This dual-pronged approach ensures that the model's foundational prior knowledge remains stable and consistent with its original state via $\mathcal{L}_{\text{DI}}$ and the finetuned model final output (the posterior) is explicitly trained for invariance across permutations via $\mathcal{L}_{\text{SI}}$.

### 3.5 THEORETICAL JUSTIFICATION

Our framework is theoretically grounded in the principle that demonstration order invariance can be formalized as enforcing the conditional independence of the model's output distribution from the permutation of demonstrations, given the underlying semantic signal. An ideal model's response should depend only on the semantic signal, which means the response should be conditionally independent of the positional noise when the signal is provided. To formalize the principle for model training, we provide Theorem 1.

**Theorem 1** (Conditional-Independence Criterion). *Let $Y, N, S$ be random variables representing the output, noise, and signal, respectively. The output $Y$ is conditionally independent of noise $N$ given signal $S$ if and only if the conditional probability distribution of the output is the same for all instances of noise. That is, for every signal $s$ and any two noise instances $n_i, n_j$:*

$$P\big(y \mid S = s, \, N = n_i\big) \;=\; P\big(y \mid S = s, \, N = n_j\big) \quad \text{for all } y. \tag{1}$$

*This implies that $Y \perp\!\!\!\perp N \mid S$, meaning $p(y \mid S = s, N = n) = p(y \mid S = s)$ for all $y, s, n$.*

Theorem 1 establishes that our objective is equivalent to training the model to produce identical output distributions for all input permutations that share the same underlying semantic signal. The proof for Theorem 1 is detailed in Appendix C.

Building on this, we provide Theorem 2 to show that our Self-Inconsistency loss, defined via the generalized JSD, provides a direct and differentiable means of realizing this objective: minimizing the JSD to zero is sufficient to guarantee distributional equivalence of model outputs, thereby enforcing the Conditional-Independence Criterion for mitigating positional noise.

**Theorem 2.** *Let an LLM be parameterized by $\theta$. For a set of $N$ input prompts $\{x_j\}_{j=1}^N$ sharing the same signal but differing in noise, if the generalized JSD between the single-step conditional output distributions $\{P_\theta(Y_t|x_j, y_{<t})\}_{j=1}^N$ is zero for all generation steps $t$, then the full generative distributions are identical. That is, if for all $t$ and all valid contexts $y_{<t}$:*

$$JSD(P_\theta(Y_t|x_1, y_{<t}), \ldots, P_\theta(Y_t|x_N, y_{<t})) = 0$$

*then for any arbitrary response sequence $y$:*

$$P_\theta(y|x_1) = P_\theta(y|x_2) = \cdots = P_\theta(y|x_N). \tag{2}$$

Therefore, minimizing our proposed losses, $\mathcal{L}_{\text{SI}}$ and $\mathcal{L}_{\text{DI}}$, drives the JSD toward zero. This satisfies the condition in Equation 2, directly enforcing the Conditional-Independence Criterion (Equation 1 of Theorem 1) and compelling the model to become invariant to the demonstration order. The proof for Theorem 2 is given in Appendix D.

## 4 EXPERIMENT

We conduct experiments to evaluate the performance of our proposed SIO framework, aiming to answer the following research questions: **RQ1:** How effectively does SIO reduce an LLM's sensitivity

Table 1: **Positional Noise Robustness Comparison of Gemma-3-4B and Qwen-3-4B Models Before and After SIO alignment.** The table compares performance metrics for Qwen3-4B and Gemma-3-4B models on three math datasets in their original pretrained state and after alignment finetuning with our SIO framework. The arrows indicate the desired direction for each metric: (↑) higher is better, (↓) lower is better. The relative change, $\Delta\%$, is calculated as $\frac{(\text{after}-\text{before})}{\text{before}} \times 100$. Green and red cells in the $\Delta\%$ columns denote performance improvement and deterioration, respectively, while gray indicates no change. The Self-Inconsistency metric is macro-averaged over all test samples from the combined datasets.

| Dataset | Eval Metric | Gemma-3-4B | | | Qwen-3-4B | | |
|---|---|---|---|---|---|---|---|
| | | Before | After | $\Delta\%$ | Before | After | $\Delta\%$ |
| AIME | Accuracy (↑) | 0.320 | 0.320 | 0.000 | 0.625 | 0.655 | 4.800 |
| | Jaccard Distance (↓) | 0.660 | 0.650 | -1.635 | 0.162 | 0.167 | 3.021 |
| | Accuracy Difference (↓) | 0.140 | 0.110 | -21.429 | 0.070 | 0.090 | 28.571 |
| GSM8K | Accuracy (↑) | 0.930 | 0.935 | 0.538 | 0.980 | 0.980 | 0.000 |
| | Jaccard Distance (↓) | 0.181 | 0.171 | -5.310 | 0.037 | 0.029 | -21.096 |
| | Accuracy Difference (↓) | 0.110 | 0.060 | -45.455 | 0.015 | 0.005 | -66.667 |
| MATH | Accuracy (↑) | 0.810 | 0.840 | 3.704 | 0.785 | 0.795 | 1.274 |
| | Jaccard Distance (↓) | 0.363 | 0.357 | -1.461 | 0.114 | 0.107 | -6.824 |
| | Accuracy Difference (↓) | 0.120 | 0.160 | 33.333 | 0.020 | 0.030 | 50.000 |
| Overall | Accuracy (↑) | 0.687 | 0.698 | 1.689 | 0.797 | 0.810 | 1.669 |
| | Jaccard Distance (↓) | 0.401 | 0.393 | -2.143 | 0.104 | 0.101 | -3.356 |
| | Accuracy Difference (↓) | 0.123 | 0.110 | -10.787 | 0.035 | 0.042 | 19.143 |
| | Self-Inconsistency (↓) | 1.528 | 1.082 | -29.150 | 4.879 | 3.901 | -20.043 |

to the order of in-context demonstrations? **RQ2:** Does this improved invariance come at the cost of task accuracy? **RQ3:** Can the framework's robustness generalize to other semantic-preserving input variations, such as tokenization noise?

## 4.1 EXPERIMENT SETUP

**Models and Datasets.** We conduct experiments on two open-source LLMs, Qwen-3-4B (Qwen Team, 2025) and Gemma-3-4B (Gemma Team, Google DeepMind, 2025). To rigorously evaluate both task performance and input positional invariance, we selected three mathematical reasoning benchmarks with progressively increasing difficulty. These include GSM8K (Cobbe et al., 2021), a dataset of grade-school math word problems; MATH (Hendrycks et al., 2021), a benchmark of challenging high-school competition problems; and AIME (Neubig, 2024), a set of prestigious and highly difficult invitational mathematics examination problems.

**Training Details.** We employed Low-Rank Adaptation (LoRA) for parameter-efficient finetuning, specifically targeting the attention blocks and the final unembedding layer (Hu et al., 2022). We believe that positional sensitivity stems from the attention mechanism's inconsistent identification of contextual information. Finetuning the attention blocks encourages a consistent focus on semantic content, regardless of positional variations. The application of LoRA to the unembedding layer implements the value network in our Bayesian update inspired SIO framework, enabling the model to learn preference distributions. In addition to LoRA, we also finetuned the weights of the normalization layers. Further training details, including hyperparameter setup and engineering optimizations, are available in Appendix E.

## 4.2 MITIGATING ICL DEMONSTRATION ORDER SENSITIVITY (RQ1)

To quantify the improvement in demonstration order invariance, we evaluated the models on our key metrics before and after applying Self-Inconsistency Optimization. As summarized in Table 1, our framework substantially reduced inconsistency for both Gemma-3-4B and Qwen-3-4B. We observed a relative decrease in Jaccard Distance of up to 21% and a reduction in Pairwise Accuracy

Table 2: A case study showing a fine-tuned Gemma-3-4b model improves answer consistency across two input permutations. The pretrained base model failed to produce at least one parsable answer for one permutation, resulting in an empty set $\varnothing$ and showed complete disagreement between contextual groups (Jaccard Distance = 1.0). The fine-tuned model resolved this, achieving perfect consensus (Jaccard Distance = 0.0) and accuracy (Absolute Accuracy Difference = 0.0).

| **Case Study of Answer Consistency Across Two Input Permutation** | | | |
|---|---|---|---|
| **Component** | Details | | |
| **Question** | Suppose $t = 5l + 384$, $2t - t + 309 = -4l$. What is the hundreds digit of $\frac{-22}{l} - \frac{3440}{-14}$? | | |
| **Ground Truth** | 2 | | |
| | **Base Model** | **Finetuned Model** | **Change ($\Delta$)** |
| **Group 0 Answers** | $\varnothing$ | $\{2.0\}$ | $+\{2.0\}$ |
| **Group 1 Answers** | $\{2.0, 4.0\}$ | $\{2.0\}$ | $-\{4.0\}$ |
| **Jaccard Distance** | 1.0 | 0.0 | **-1.0** |
| **Abs. Acc. Diff.** | 1.0 | 0.0 | **-1.0** |

Difference by as much as 66%, indicating that responses generated from different permutations became substantially more aligned. While most metrics improved, a slight increase in accuracy difference on MATH suggests that achieving invariance on more difficult numerical reasoning tasks may require further tuning. The consistent downward trend of the loss curves in Figure 4, which had not yet fully converged within our compute budget, suggests that further improvements are possible with additional training. Taken together, these results show that **models finetuned with our SIO framework converge towards a consensus response set that consistently includes the correct answer, effectively neutralizing the influence of in-context example order.**

### 4.3 PRESERVING TASK ACCURACY (RQ2)

The improved invariance from SIO is achieved without compromising, and in some cases even improving, task accuracy. As shown in Table 1, post-finetuning performance did not degrade. Instead, both models showed an average relative accuracy increase of over 1.6% across all datasets. Notably, Gemma-3-4B's accuracy on the challenging AIME benchmark increased by 4.8%. The stability of the accuracy rewards during rollout, illustrated in Figure 3, further indicates that the model does not sacrifice correctness for consistency. These results provide strong evidence that **SIO induces robustness while maintaining or enhancing the model's core problem-solving capabilities.**

### 4.4 CASE STUDY

Beyond aggregated statistics, the case study in Table 2 provides granular evidence of our method's effectiveness. The example compares outputs from two inputs with identical content but different demonstration orders. The base model was highly unstable: it failed to produce a valid answer for one permutation while generating two different answers for the other, resulting in maximal inconsistency (Jaccard Distance of 1.0). In stark contrast, the SIO-finetuned model achieved perfect consistency, converging to the single correct answer for both permutations and yielding ideal scores of 0.0 for both Jaccard Distance and Absolute Accuracy Difference. This case study provides a concrete example of how **our method compels the model to produce identical, accurate responses to semantically equivalent inputs, neutralizing the influence of demonstration order.**

### 4.5 GENERALIZING TO TOKENIZATION NOISE (RQ3)

To assess whether our framework's robustness extends beyond demonstration order, we evaluated its performance against tokenization noise. We augmented the training data with two types of semantic-

Table 3: Robustness Comparison of Gemma-3-4B and Qwen-3-4B Models Before and After SIO Alignment Across Various Noise Types. The table compares performance metrics for Qwen3-4B and Gemma-3-4B models on the GSM8K dataset in their original pretrained state and after alignment finetuning with our SIO framework. The arrows indicate the desired direction for each metric: ($\uparrow$) higher is better, ($\downarrow$) lower is better. The relative change in percentage, $\Delta\%$, is calculated as $\frac{\text{(after}-\text{before)}}{\text{before}} \times 100$. Green and red cells in the $\Delta\%$ columns denote performance improvement and deterioration, respectively, while gray indicates no change.

| Noise Type | Eval Metric | Gemma-3-4B | | | Qwen-3-4B | | |
|---|---|---|---|---|---|---|---|
| | | Before | After | $\Delta\%$ | Before | After | $\Delta\%$ |
| **Order** | Accuracy ($\uparrow$) | 0.930 | 0.935 | 0.538 | 0.980 | 0.980 | 0.000 |
| | Jaccard Distance ($\downarrow$) | 0.181 | 0.171 | -5.310 | 0.037 | 0.029 | -21.096 |
| | Accuracy Difference ($\downarrow$) | 0.110 | 0.060 | -45.455 | 0.015 | 0.005 | -66.667 |
| | Self-Inconsistency ($\downarrow$) | 1.528 | 1.082 | -29.150 | 4.879 | 3.901 | -20.043 |
| **Case** | Accuracy ($\uparrow$) | 0.930 | 0.950 | 2.151 | 0.980 | 0.980 | 0.000 |
| | Jaccard Distance ($\downarrow$) | 0.182 | 0.174 | -4.879 | 0.027 | 0.026 | -4.494 |
| | Accuracy Difference ($\downarrow$) | 0.100 | 0.095 | -5.000 | 0.005 | 0.005 | 0.000 |
| | Self-Inconsistency ($\downarrow$) | 11948.600 | 4177.870 | -65.035 | 45476.500 | 17317.700 | -61.919 |
| **Typo** | Accuracy ($\uparrow$) | 0.860 | 0.930 | 8.140 | 0.980 | 0.980 | 0.000 |
| | Jaccard Distance ($\downarrow$) | 0.207 | 0.185 | -10.859 | 0.021 | 0.023 | 10.096 |
| | Accuracy Difference ($\downarrow$) | 0.115 | 0.105 | -8.696 | 0.010 | 0.005 | -50.000 |
| | Self-Inconsistency ($\downarrow$) | 11653.200 | 4058.750 | -65.171 | 44108.600 | 16635.700 | -62.285 |
| **Order + Case + Typo** | Accuracy ($\uparrow$) | 0.910 | 0.940 | 3.297 | 0.980 | 0.980 | 0.000 |
| | Jaccard Distance ($\downarrow$) | 0.329 | 0.206 | -37.462 | 0.037 | 0.033 | -11.444 |
| | Accuracy Difference ($\downarrow$) | 0.225 | 0.140 | -37.778 | 0.010 | 0.010 | 0.000 |
| | Self-Inconsistency ($\downarrow$) | 11950.000 | 4211.080 | -64.761 | 45205.500 | 17136.600 | -62.092 |

preserving perturbations designed to simulate real-world input variations: **Random Space Insertion** (adding extraneous whitespace) and **Random Case Conversion** (altering text capitalization). These variations introduce significant token-level differences without altering the underlying meaning. Our experiments on the GSM8K dataset, with results summarized in Table 3, demonstrate a direct correlation between input variation and output inconsistency. The addition of tokenization noise, combined with demonstration order permutation, increased the self-inconsistency score by nearly four orders of magnitude compared to order permutation alone. Despite this extreme input variance, SIO proved highly effective, reducing the Self-Inconsistency score by over 60%. The model's ability to stably optimize against such a large loss (see loss curve analysis in Appendix G) further underscores the robustness of our training architecture. These findings confirm that **SIO is a generalizable framework for improving LLM robustness, effectively mitigating sensitivity to superficial input variations beyond demonstration order.**

## 5 CONCLUSION

In this work, we address the critical issue of LLM sensitivity to the order of in-context demonstrations. To mitigate this reliability gap, we introduce Self-Inconsistency Optimization (SIO), a simple model agnostic post-training framework that enhances robustness to superficial input variations. Our method encourages the model to prioritize semantic meaning over positional noise by minimizing the Jensen–Shannon divergence across semantically equivalent prompts.

Theoretically, we establish that reducing self-inconsistency to zero enforces order invariance. To achieve this in practice without compromising task performance, we propose a Bayesian update-inspired architecture. This design, implemented as a product-of-experts with a frozen prior model and a lightweight adapter, effectively separates knowledge retention from preference alignment, leading to more stable optimization. Empirically, we demonstrate that SIO consistently reduces distributional instability on the GSM8K, MATH, and AIME benchmarks while maintaining or improving accuracy. These benefits extend beyond demonstration order to other variations like tokenization noise. Ultimately, SIO offers a general, theoretically grounded, and practical approach to developing more robust and trustworthy LLMs, marking a significant step toward their reliable deployment in high-stakes, real-world applications.

ETHICS STATEMENT

This research does not involve human subjects, personally identifiable data, or sensitive attributes. All datasets used are publicly available benchmark datasets that have been widely adopted in prior work. We are not aware of any potential harm associated with releasing our methods, as with any AI technology, misuse could occur if applied irresponsibly in high-stakes settings. We encourage careful consideration of societal impacts and safe deployment practices.

REPRODUCIBILITY STATEMENT

We follow ICLR's reproducibility guidelines. All datasets used in this study are publicly available, and we provide detailed dataset descriptions. The proposed method is implemented using standard libraries, and we have provided the codebase, including training scripts, hyperparameter settings, and evaluation procedures for paper review purposes.

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

APPENDIX

## A    USE OF LLMS

We used a large language model only for spelling and grammar correction of the manuscript text. The LLM was not involved in research ideation, experimental design, data generation, analysis, or substantive writing beyond copy-editing. All content and claims were authored and verified by the authors, who take full responsibility for the paper. The LLM is not an author.

## B    THEORETICAL JUSTIFICATION FOR THE BAYESIAN FINETUNING ARCHITECTURE

Our goal is to show that combining the logits of a frozen "policy network" and a trainable "value network" is equivalent to performing a Bayesian update on their respective probability distributions.

The foundation of our argument rests on two key points: (1) the relationship between logits and log-probabilities, and (2) the properties of logarithms that translate addition into multiplication.

**Step 1: Logits as Unnormalized Log-Probabilities.**    For a given vocabulary, the probability of a token $t$ is calculated by applying the softmax function to its corresponding logit, $z_t$:

$$P(t) = \frac{\exp(z_t)}{\sum_i \exp(z_i)}$$

Taking the logarithm of both sides, we get:

$$\log P(t) = \log\left(\frac{\exp(z_t)}{\sum_i \exp(z_i)}\right) = z_t - \log\left(\sum_i \exp(z_i)\right)$$

The term $\log(\sum_i \exp(z_i))$ is the log-partition function, which is a normalization constant that is the same for all tokens in the distribution. Let's denote this constant as $C$. Thus, we can express the logit as:

$$z_t = \log P(t) + C$$

This shows that a logit vector is equivalent to the unnormalized log-probabilities of the tokens.

**Step 2: Mapping Model Components to Bayesian Terms.**    We map the components of our architecture to the terms in Bayes' theorem, $P(\text{posterior}) \propto P(\text{likelihood}) \times P(\text{prior})$:

- **Prior, $P_{\textbf{prior}}(t)$:** The probability distribution produced by the frozen, pretrained policy network. Its logits are $z_{\text{policy}}$.
- **Likelihood, $P_{\textbf{likelihood}}(t)$:** The probability distribution implicitly modeled by the trainable LoRA value network. Its logits are $z_{\text{value}}$.
- **Posterior, $P_{\textbf{posterior}}(t)$:** The final, aligned probability distribution from the combined model. Its logits are $z_{\text{final}}$.

**Step 3: Derivation from Additive Logits to Multiplicative Probabilities.**    Our model architecture combines the logits additively:

$$z_{\text{final}} = z_{\text{policy}} + z_{\text{value}}$$

Using the relationship from Step 1, we can substitute the log-probability expressions for each logit term:

$$\log P_{\text{posterior}}(t) + C_{\text{post}} = (\log P_{\text{prior}}(t) + C_{\text{prior}}) + (\log P_{\text{likelihood}}(t) + C_{\text{like}})$$

We can group all the normalization constants into a single new constant, $C' = C_{\text{prior}} + C_{\text{like}} - C_{\text{post}}$:

$$\log P_{\text{posterior}}(t) = \log P_{\text{prior}}(t) + \log P_{\text{likelihood}}(t) + C'$$

Using the logarithm property that $\log(A) + \log(B) = \log(A \times B)$, we get:

$$\log P_{\text{posterior}}(t) = \log(P_{\text{prior}}(t) \times P_{\text{likelihood}}(t)) + C'$$

Exponentiating both sides removes the logarithms:

$$\exp(\log P_{\text{posterior}}(t)) = \exp(\log(P_{\text{prior}}(t) \times P_{\text{likelihood}}(t)) + C')$$

$$P_{\text{posterior}}(t) = \exp(C') \times (P_{\text{prior}}(t) \times P_{\text{likelihood}}(t))$$

Since $\exp(C')$ is a constant, this equation demonstrates the proportionality at the heart of Bayes' theorem:

$$P_{\text{posterior}}(t) \propto P_{\text{prior}}(t) \times P_{\text{likelihood}}(t)$$

This derivation formally proves that our architecture of adding the logits from a frozen policy network and a trainable value network is not an ad-hoc engineering choice but a direct and principled implementation of a Bayesian update.

## C   PROOF OF THEOREM 1

To prove the theorem, we first fix $s$ with $\Pr(S = s) > 0$. By hypothesis equation 1, the conditional probability $P(y \mid S = s, N = n)$ is constant with respect to $n$; let this constant be $f_s(y)$. By the law of total probability, we can marginalize out $N$:

$$P(y \mid S = s) = \sum_n P(y \mid S = s, N = n) \, P(n \mid S = s)$$

$$= \sum_n f_s(y) \, P(n \mid S = s)$$

$$= f_s(y) \sum_n P(n \mid S = s)$$

$$= f_s(y).$$

This shows that $P(y \mid S = s) = f_s(y) = P(y \mid S = s, N = n)$ for all $y$ and $n$, which is the definition of conditional independence $Y \perp\!\!\!\perp N \mid S$.

## D   PROOF OF THEOREM 2

To prove the theorem, we first establish a standard lemma regarding the properties of the generalized JSD.

**Lemma 1.** *For any set of probability distributions $\{P_j\}_{j=1}^N$ and strictly positive weights $w_j > 0$ summing to one, $\text{JSD}_w(P_1, \ldots, P_N) = 0$ if and only if $P_1 = P_2 = \cdots = P_N$.*

*Proof.* The generalized JSD is defined as $\text{JSD}_w(P_1, \ldots, P_N) = \sum_{j=1}^N w_j \, \text{D}_{\text{KL}}(P_j \| M)$, where $M = \sum_{j=1}^N w_j P_j$ is the mixture distribution.

By Gibbs' inequality, the Kullback-Leibler (KL) divergence $\text{D}_{\text{KL}}(P \| Q)$ is non-negative and equals zero if and only if $P = Q$. Since the weights $w_j$ are strictly positive, each term $w_j \, \text{D}_{\text{KL}}(P_j \| M)$ in the JSD sum is non-negative.

The sum is zero if and only if every term is zero. This requires $\text{D}_{\text{KL}}(P_j \| M) = 0$ for all $j \in \{1, \ldots, N\}$. This condition holds if and only if $P_j = M$ for all $j$. Consequently, all distributions must be identical: $P_1 = P_2 = \cdots = P_N$. $\qquad\square$

We now prove Theorem 2 by induction on the generated sequence length. Our goal is to show that for any sequence $y = (y_1, \ldots, y_L)$, the probability $P_\theta(y \mid x_j)$ is constant for all $j \in \{1, \ldots, N\}$.

*Proof of Theorem 2.* We proceed by induction on the sequence length $t$, from 1 to $L_{max}$.

**Base Case ($t = 1$):**   For a sequence of length 1, the theorem's premise states that $\text{JSD}_w(\{P_\theta(Y_1 \mid x_j)\}_{j=1}^N) = 0$. By Lemma 1, this implies that the next-token distributions are identical:

$$P_\theta(Y_1 \mid x_1) = P_\theta(Y_1 \mid x_2) = \cdots = P_\theta(Y_1 \mid x_N).$$

Therefore, for any specific token $y_1$, we have $P_\theta(y_1 \mid x_1) = \cdots = P_\theta(y_1 \mid x_N)$. The base case holds.

**Inductive Hypothesis:** Assume the claim holds for all prefixes of length $t - 1$. That is, for any sequence $y_{<t} = (y_1, \ldots, y_{t-1})$, the probability of generating it is constant across all inputs:

$$P_\theta(y_{<t} \mid x_1) = P_\theta(y_{<t} \mid x_2) = \cdots = P_\theta(y_{<t} \mid x_N).$$

**Inductive Step:** We show that the claim holds for sequences of length $t$. The probability of generating a sequence $y_{\leq t} = (y_1, \ldots, y_t)$ given an input $x_j$ can be decomposed using the chain rule of probability:

$$P_\theta(y_{\leq t} \mid x_j) = P_\theta(y_{<t} \mid x_j) \cdot P_\theta(y_t \mid x_j, y_{<t}).$$

By the inductive hypothesis, the first term, the prefix probability $P_\theta(y_{<t} \mid x_j)$, is constant for all $j$.

For the second term, the theorem's premise states that for any prefix $y_{<t}$, the JSD of the next-token distributions is zero:

$$\mathrm{JSD}_w(\{P_\theta(Y_t \mid x_j, y_{<t})\}_{j=1}^N) = 0.$$

Applying Lemma 1, it follows that these conditional distributions are identical. Thus, for our specific token $y_t$:

$$P_\theta(y_t \mid x_1, y_{<t}) = P_\theta(y_t \mid x_2, y_{<t}) = \cdots = P_\theta(y_t \mid x_N, y_{<t}).$$

Since both the prefix probability and the conditional next-token probability are constant across all $j$, their product, $P_\theta(y_{\leq t} \mid x_j)$, must also be constant.

This completes the inductive step. Therefore, the theorem holds for any sequence length. $\square$

# E  TRAINING DETAILS

## E.1  REWARD-ALIGNED CONSISTENCY LOSS

To enhance training stability and convergence speed in our preference learning framework, we introduce a **reward-aligned weighting scheme** for computing the weight $w_j$ in the mixture distribution of the consistency loss. Instead of a uniform average, this method strategically assigns weights based on the reward signal. Specifically, it gives greater weight to low-probability responses that receive a negative reward and to high-probability responses that receive a positive reward. This alignment encourages the model to more strongly penalize undesirable, unlikely outputs while reinforcing desirable, confident predictions, which helps to speed up the convergence of the preference learning.

## E.2  PIPELINE IMPLEMENTATION

Our implementation leverages the **vLLM library** for efficient LLM inference and the **Hugging Face Accelerate library** for fine-tuning.

**Training Data Generation.**  Each dataset is preprocessed by partitioning it into four disjoint subsets: demonstration, training, validation, and test. The training subset contains the target questions the model must answer, while the demonstration subset provides the in-context examples. The validation subset is used to monitor training performance (e.g., effectiveness and stability), and the test subset is reserved to assess final model performance. For each training iteration, we constructed a rich training dataset from 36 questions of each math dataset, each augmented with 8 in-context learning examples. For each question of interest, we created two distinct augmented inputs and sampled 14 responses with temperature ranged between 0.8 and 1.9 for each, resulting in 28 responses per question. To further improve learning, we incorporated a replay buffer by adding 4 distinct, known-correct answers to each set of rollout. We set max response length as 4096 tokens and discarded the responses that exceeds this limit. Therefore, the total training dataset contains 108 questions and max of 3456 responses. Two reward signals guided the training process: a formatting reward to encourage the model to wrap the final numerical answer for straightforward extraction, and an accuracy reward based on the correctness of the extracted answer when compared to the ground-truth solution.

**LLM alignment finetuning.**  For parameter-efficient optimization, we employed Stable LoRA with a rank of 32. The entire pipeline, comprising data generation (rollout) and training, was executed for 20 iterations on 3 NVIDIA A100 GPUs, with a total training time of approximately 3 days. For each iteration, we used the AdamW optimizer with a weight decay of 0.1 and a learning

rate of $3 \times 10^{-7}$, managed by a cosine scheduler with a 20% warm-up ratio. The model was trained for 8 epochs with an effective batch size of 96. The final objective loss hyperparameters are $\lambda_1 = 1, \lambda_2 = 1, \lambda_3 = 0.01, \lambda_4 = 5 \times 10^{-6}$. The clipping epsilon for GSPO surrogate loss is using the default from the original paper as setting the left and right clipping ranges to 3e-4 and 4e-4, respectively.

**Evaluation Protocol.** For evaluation, we used a separate set of 200 questions from each math dataset, also presented with 8 in-context learning examples. To simulate a practical use case that balances generation speed with quality, we sampled 8 responses with temperature of 0.5 for each augmented input during evaluation .

## F  RELATED WORK

LLMs often change their answers when the same set of demonstration examples is merely reordered. This behavior indicates a dependence on superficial positional cues in the prompt rather than on the underlying semantic content, which undermines reliability—especially in high-stakes settings where consistent reasoning is required. A desirable property, therefore, is order invariance: predictions should be stable when semantically equivalent demonstrations are rearranged. We adopt this reliability lens and focus the discussion on how prior work attempts to reduce order sensitivity and where those attempts fall short.

**Prior approaches and their limitations.** Existing methods fall into three broad families. First, inference-time adjustments keep model weights fixed and manipulate the prompt or decoding. Typical strategies include searching for a high-performing ordering of demonstrations, templating and reformatting prompts, and ensembling multiple responses from different permutations (Lu et al., 2022; Wang et al., 2023; He et al.; Zhao et al., 2021; Zhang et al., 2024). These techniques can lift accuracy without retraining, but they add test-time cost and latency, and they primarily steer around the superficial noise by finding a "good" order rather than removing the underlying sensitivity. They also offer no guarantee that predictions will remain stable outside the searched permutations. Second, training-time optimization finetunes models to be less order-sensitive, for example by enforcing agreement on designated hidden states across augmented prompts, distilling from teachers that average over permutations, or using distributionally robust and adversarial schedules that expose difficult orderings (Chen et al., 2025; Xiang et al., 2024; Liusie et al., 2024). While effective, these approaches often rely on specialized teachers or inner optimization loops that increase engineering complexity and compute. Guarantees are empirical only and tied to the specific augmentations seen during training, and many evaluations emphasize multiple-choice settings, leaving complex generative reasoning less explored. Third, architectural modifications change how demonstrations are processed by treating them as a set, altering attention patterns, or adding invariant aggregation modules (Egressy & Stühmer, 2025; Fang et al., 2025). Such designs can deliver strong invariance on paper but require non-standard implementations that complicate deployment and may interact unpredictably with tasks where order is genuinely meaningful. *Shared limitations across families* we target are: **L1** increased test-time compute/latency (search or ensembling), **L2** lack of formal invariance guarantees beyond observed permutations, **L3** dependence on non-standard sophisticated implementations or training design with significant computational overhead, and **L4** evaluations that do not stress generative tasks like complex mathematical reasoning.

**Our approach and how it addresses these limitations.** SIO is a *drop-in architecture-agnostic* post-training framework that enforces distributional *self-alignment across permutations*: for permutation-equivalent prompts, we minimize a Jensen–Shannon divergence between the model's output distributions. This directly *removes* nuisance positional dependence instead of searching for or ensembling "good" orders, and integrates seamlessly with existing post-training pipelines (including RL with a Bayesian-inspired update) without modifying the backbone or decoding stack to improve training stability and efficiency. Concretely, SIO addresses the shared limitations as follows: (**L1**) *No extra test-time cost*: invariance is learned during training, so inference uses a single pass without ordering search or multi-sample aggregation; (**L2**) *Formal guarantee*: we prove that minimizing the self-inconsistency loss suffices to achieve invariance over the targeted permutation group (see §3.5); (**L3**) *Standard implementation*: no teacher models, min–max loops, or bespoke attention are required as only an additive JS regularizer over augmented prompts is needed, hence making the method deployment-friendly; (**L4**) *Stronger evaluation*: we evaluate on *generative* mathematical

reasoning tasks (beyond multiple-choice), demonstrating stability under complex generation. Taken together, SIO balances practicality (drop-in, low engineering overhead) with principled robustness (distributional alignment and a formal invariance justification), directly targeting the core failure mode identified above.

## G SELF-INCONSISTENCY LOSS ANALYSIS

Since the JSD-based Self-Inconsistency score serves as both an evaluation metric and a training objective, its trend offers insight into the framework's stability and effectiveness. As the tokenization noise experiment demonstrated, greater input deviation leads to greater output inconsistency. Even when faced with an evaluation loss three orders of magnitude larger—resulting from combined position, typo, and case noise—than that of positional noise alone, our framework stably optimized the LLM toward a consistent output distribution.

The loss curves in Figure 4 confirm the stability of our Bayesian update-inspired training architecture. Both training and evaluation losses exhibit a steady downward trend, indicating that the model effectively and smoothly aligns its output distributions across permutations. Crucially, we observed no instances of model collapse or reward degradation during training. This stability underscores the robustness of the Bayesian update framework, which successfully guides the model toward a consistent and accurate state without the common pitfalls, such as training instability, associated with preference alignment.

## H    EXPERIMENTAL RESULT PLOTS

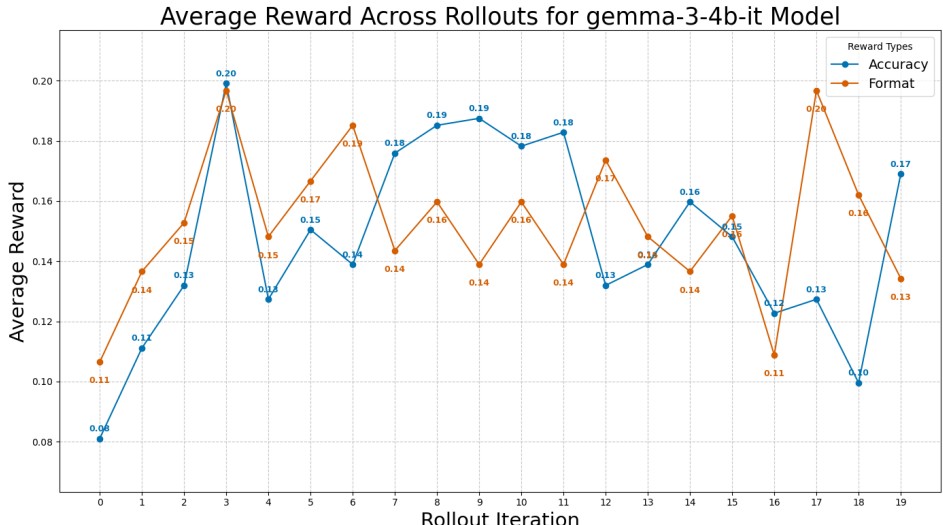

(a) Gemma-3-4b Model Reward Plot

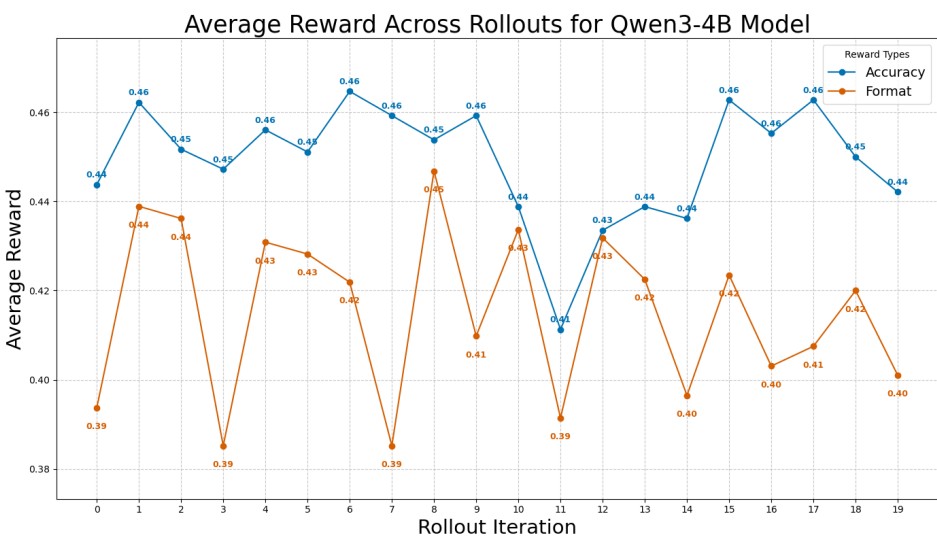

(b) Qwen-3-4b Model Reward Plot

Figure 3: Average accuracy and format rewards across 20 rollout iterations for the (a) Gemma-3-4b-it and (b) Qwen3-4B models. The absence of a consistent downward trend in the reward curves indicates that the performance of neither model degraded during the rollout process.

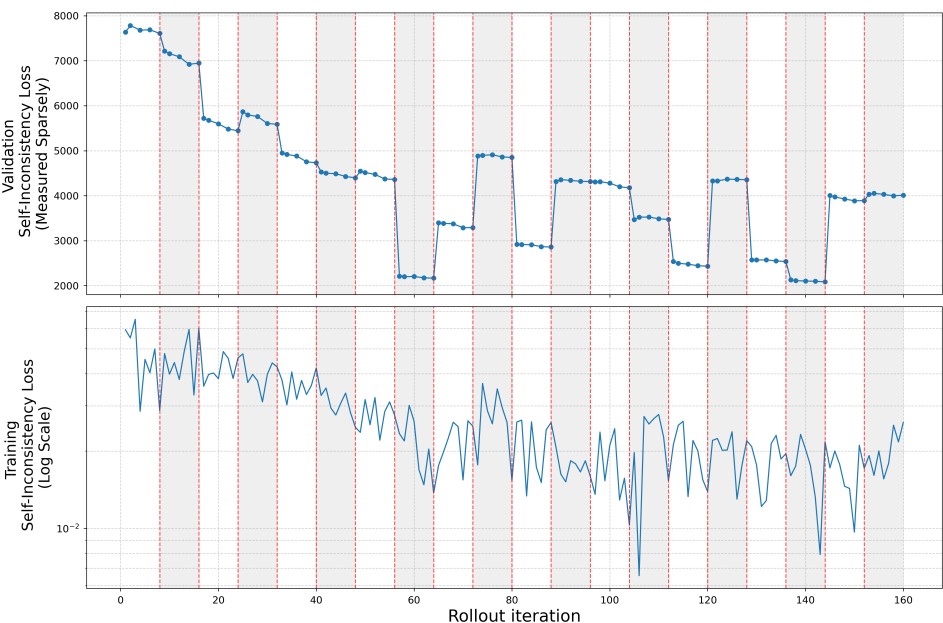

(a) Loss for Gemma-3-4b Model

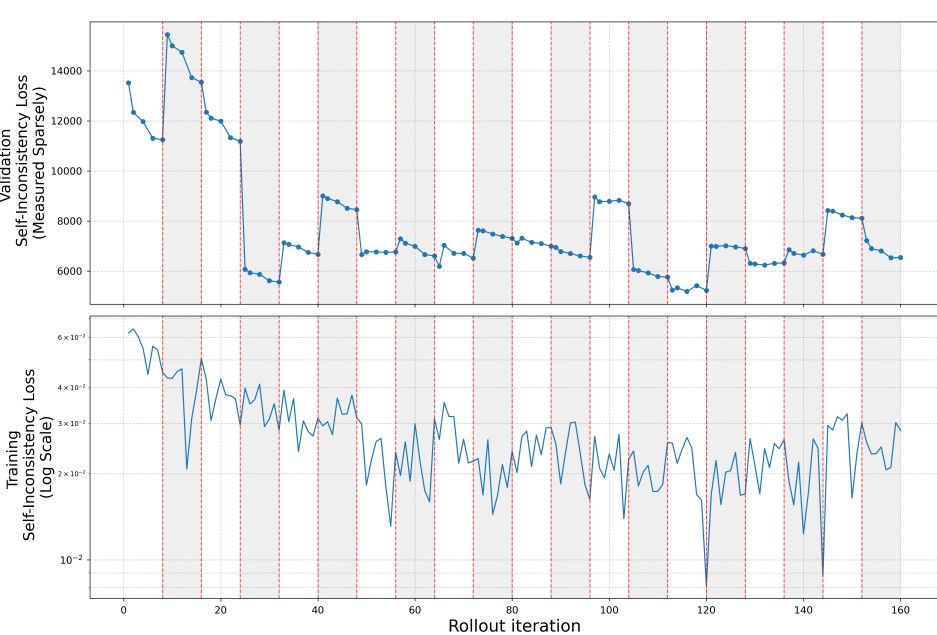

(b) Loss for Qwen-3-4B Model

Figure 4: Self-inconsistency loss curves for training and validation of the (a) Gemma-3-4b and (b) Qwen-3-4B models. For each model, the top panel displays the validation loss, measured sparsely, and the bottom panel displays the training loss (log scale) over rollout iterations. The shaded regions mark the beginning of each new rollout with augmented data. A general downward trend in the validation loss for both models indicates successful learning and improvement.

