# OpenReview forum: "From Self-Inconsistency to Stability: Achieving Order Invariant In-Context Learning"
_ICLR.cc/2026/Conference — ICLR 2026 Conference Withdrawn Submission_

### Official Review · Reviewer_oq8K · 2025-10-25

**Soundness:** 2
**Presentation:** 4
**Contribution:** 2
**Rating:** 2
**Confidence:** 3

**Summary:**

The authors introduce a new training scheme for encouraging permutation invariance to exemplar order in ICL. They demonstrate that their methodology appears to work on a math dataset.

**Strengths:**

The paper is clear and well written. The topic is interesting and timely. The proposed training methodology sounds interesting, and seems to have some potential.

**Weaknesses:**

The differences made by the training methodology seem to be small. Looking at your final results in Table 1, accuracy differences seem to improve overall by 0.01 for Gemma-3-4B, and appear to actually *worsen* for Qwen-3-4B. The magnitude of accuracy difference improvement seems to be fairly small, so comparing percentage change appear to inflate these differences. Improvement is also inconsistent across models and tasks. Hence, it's difficult to judge how significant of a difference your methodology is actually making. Were you able to replicate your results over multiple seeds, to gain a sense of the degree in variation?

It's also unclear whether your methodology is necessary at all. Looking at the Qwen column, for example, accuracy difference is already just 0.035 overall (and even smaller in Table 3). Qwen also seems to perform markedly better than Gemma overall. Perhaps a stronger model already learns a degree of permutation invariance, and your specific method is unnecessary?

It was also unclear which components of your pipeline were necessary. Were you able to perform ablation studies to see whether simple data augmentation is enough to reproduce your results? Are the regularization terms necessary in your loss? It also appears that your Bayesian update explanation (Section 3.4) seems to be a simple re-interpretation of LoRA training, which in turn is equivalent to vanilla finetuning if the LoRA rank is equal to the base weight rank. If vanilla finetuning can be interpreted this way, is this a useful construct?

The theory section (Section 3.5) seems to be a little extraneous. The conclusions are perhaps sufficiently intuitive/obvious that a formal argument is unnecessary. Perhaps the extra space could be more efficiently used for ablation results / additional empirics with more models and more tasks?

**Questions:**

See Weaknesses above.

---

> ### Author Response · Authors · 2025-12-03
>
> [W1] **Magnitude of Improvement and Reproducibility:** We addressed the concern regarding small margins and variance by rerunning our experiments with a new random seed and significantly increasing the sample size to 1,122 questions (10x larger) with 16 demonstrations ($m=16$).
>
> Our results (Table 2) confirm that SIO consistently improves stability. We emphasize that the primary goal of SIO is to enforce **permutation invariance** (reducing sensitivity to noise) while **preserving** task accuracy. We are not aiming for SOTA accuracy gains; rather, we aim to ensure the model does not degrade (aligning to wrong answers) while achieving stability. The fact that SIO maintains or slightly improves accuracy while significantly reducing inconsistency satisfies this research objective.
>
>
> [W2] **Necessity on Stronger Models (Qwen Scaling):** To test if stronger models naturally learn permutation invariance, we benchmarked Qwen 3 at multiple scales (4B, 8B, 14B, 32B). As shown in Table 1 below, scaling up does not linearly solve the inconsistency problem. For instance, Qwen3-32B exhibits the same Mean Abs Acc Diff as the 8B model (0.04) even if it has better accuracy. This non-linear behavior demonstrates that model scale alone is insufficient to guarantee robustness against permutation noise, confirming the necessity of a dedicated method like SIO even for larger models.
>
>
> Table 1: Result for stronger Qwen 3 models at different scale.
> | Dataset   | Eval Metric           | Qwen3-4B | Qwen3-8B | Qwen3-14B | Qwen3-32B |
> | :--- | :--- | :---: | :---: | :---: | :---: |
> | **AIME** | Pass@k Accuracy (↑)   |   0.66   |   0.63   |   0.71    |   0.71    |
> |           | Mean Jaccard Dist (↓) |   0.15   |   0.23   |   0.15    |   0.11    |
> |           | Mean Abs Acc Diff (↓) |   0.08   |   0.08   |   0.08    |   0.07    |
> | **GSM8K** | Pass@k Accuracy (↑)   |   0.97   |   0.98   |   0.98    |   0.98    |
> |           | Mean Jaccard Dist (↓) |   0.03   |   0.04   |   0.03    |   0.02    |
> |           | Mean Abs Acc Diff (↓) |   0.02   |   0.01   |   0.01    |   0.01    |
> | **MATH** | Pass@k Accuracy (↑)   |   0.84   |   0.81   |   0.81    |   0.85    |
> |           | Mean Jaccard Dist (↓) |   0.11   |   0.14   |   0.09    |   0.13    |
> |           | Mean Abs Acc Diff (↓) |   0.05   |   0.03   |   0.02    |   0.03    |
> | **Overall**| Pass@k Accuracy (↑)  |   0.82   |   0.80   |   0.83    |   0.85    |
> |           | Mean Jaccard Dist (↓) |   0.10   |   0.14   |   0.09    |   0.09    |
> |           | Mean Abs Acc Diff (↓) |   0.05   |   0.04   |   0.04    |   0.04    |

---

> ### Author Response · Authors · 2025-12-03
>
> [W3] **Component Necessity and Bayesian Interpretation:** We performed a comprehensive ablation study (Table 2) to identify necessary components.
>
> * Simplification: We found that Group Sequence Policy Optimization (GSPO) and Distributional Inconsistency (DI) loss can be removed without hurting performance. In fact, the "No DI Loss" variant yielded the best results.
> * Bayesian Architecture: Regarding the Bayesian update, while it shares implementation similarities with LoRA, the critical distinction is the architectural constraint: strictly separating the frozen prior (pretrained head) from the trainable likelihood (adapter head).
>     * The "No Bayesian" row in Table 2 represents standard fine-tuning (where this constraint is removed).
>     * The Bayesian architecture outperforms the "No Bayesian" approach.
>     * This confirms that the structural enforcement of the prior is a useful and necessary construct for stability, even when the explicit DI loss term is removed.
>
> Based on these findings, we have streamlined the SIO pipeline. The final proposed method relies solely on the Self-Inconsistency (SI) loss with LoRA applied to the transformer blocks and unembedding layer, removing both GSPO and DI losses.
>
> Table 2: Ablation results with 1122 questions and 16 demonstrations.
> | Variant | AIME Acc ($\uparrow$) | AIME Jaccard ($\downarrow$) | AIME Diff ($\downarrow$) | GSM8K Acc ($\uparrow$) | GSM8K Jaccard ($\downarrow$) | GSM8K Diff ($\downarrow$) | MATH Acc ($\uparrow$) | MATH Jaccard ($\downarrow$) | MATH Diff ($\downarrow$) | Overall Acc ($\uparrow$) | Overall Jaccard ($\downarrow$) | Overall Diff ($\downarrow$) |
> | :--- | :---: | :---: | :---: | :---: | :---: | :---: | :---: | :---: | :---: | :---: | :---: | :---: |
> | **Base Model** | 0.334 | 0.661 | 0.110 | 0.925 | 0.218 | 0.099 | 0.818 | 0.438 | 0.166 | 0.693 | 0.439 | 0.125 |
> | **Standard SIO** | **0.353** | 0.665 | 0.131 | 0.914 | 0.216 | 0.088 | 0.816 | 0.464 | 0.158 | 0.694 | 0.448 | 0.126 |
> | **No GSPO** | 0.321 | 0.652 | 0.094 | **0.936** | 0.217 | 0.102 | 0.840 | 0.431 | 0.152 | 0.699 | 0.433 | 0.116 |
> | **No DI Loss** | 0.332 | 0.663 | 0.099 | 0.933 | **0.160** | **0.043** | **0.842** | **0.387** | **0.131** | **0.702** | **0.403** | **0.091** |
> | **No SI Loss** | 0.342 | 0.658 | 0.102 | 0.933 | 0.211 | 0.099 | 0.805 | 0.444 | 0.139 | 0.693 | 0.438 | 0.113 |
> | **No Bayesian** | 0.332 | 0.652 | **0.091** | 0.928 | 0.211 | 0.078 | 0.832 | 0.429 | 0.136 | 0.697 | 0.431 | 0.102 |
> | **Uniform Weights** | 0.321 | **0.648** | 0.096 | 0.928 | 0.218 | 0.096 | 0.824 | 0.443 | 0.150 | 0.691 | 0.436 | 0.114 |
>
>
>
> [W4] **Value of Theoretical Formalization:** We appreciate that reviewer finds our proof is clear enough to sound intuitive and trivial, but we believe that formalizing the intuition into a mathematical guarantee (Theorem 1 & 2) is a necessary step to transform a heuristic engineering effort into a principled framework. The theory proves why minimizing JSD is sufficient for invariance, providing a guarantee that empirical observation alone cannot supply. Also, once the theoretical guarantee is obtained, the complexity of the problem is reduced to a simple engineering work by aligning with the theory.

---

### Official Review · Reviewer_Vc7q · 2025-10-30

**Soundness:** 1
**Presentation:** 2
**Contribution:** 1
**Rating:** 2
**Confidence:** 4

**Summary:**

The authors argue that the ICL performance of today's LLMs is incomplete because they lack robustness to the order of demonstrations. While prior research has addressed this issue, the authors focus specifically on reasoning models that face the same challenge. They propose a method called Self-Inconsistency Optimization (SIO), a post-training framework that makes models robust to the order of demonstrations.

**Strengths:**

- The problem that the authors address is important. The model's lack of robustness to the order of examples naturally raises the question of whether it truly understands the context or merely memorises patterns.
- The method demonstrates significant improvements in robustness on both benchmark datasets and simulated real-world input variations.

**Weaknesses:**

- The paper does not demonstrate the inconsistency of LLMs.
- There is no baseline against which the suggested method can be compared. The paper only shows the SIO's evaluation results. It is unclear whether the results are good or bad.
- There is no ablation study at all. The authors need to conduct experiments to verify the effectiveness of:
    1) Jensen-Shannon divergence, especially how to set $w_i$.
    2) Bayesian update: It is convincing, but evidence is needed.
    3) DI & SI loss
- The authors explain that the increase in accuracy was due to underfitting.

**Questions:**

1. How big is the variance when the demonstrations are randomly shuffled?
2. Would using a different RL framework (e.g., GRPO) mitigate the inconsistency problem?
3. As far as I know, the MATH dataset is very small. Can you fully train your model with SIO to reduce the "Accuracy Difference" on MATH data?

---

> ### Author Response · Authors · 2025-12-03
>
> [W1] **Demonstrating Inconsistency:** We respectfully disagree with the premise that inconsistency was not demonstrated. Our metrics (Table 1 & 3) quantify this explicitly: any Jaccard Distance or Accuracy Difference $>0$ proves the model is sensitive to noise.
>
> Quantitatively, when all three noise types are present for Gemma 3, the Accuracy Difference is 0.225. This indicates that for 22.5% of questions, the model flips between a correct and incorrect answer solely due to input permutation, despite the semantic content remaining identical (Please see Table 2 in our paper for a concrete case study). This is a significant reliability failure (approx. 1 in 5 queries) that our paper aims to solve.
>
>
>
> [W2] **Baseline Comparisons:** We addressed the lack of baselines by aligning our evaluation strictly with the PEARL (ICLR 2025) benchmark protocols. We applied SIO to the PEARL tasks (CSQA, CurDial, CoLA, TMW) using their exact settings.
>
> As shown in Table 1 below, SIO outperforms PEARL, achieving top-1 or top-2 performance in 91% of metrics (41/45). Crucially, SIO demonstrates superior stability, achieving a significantly smaller "Gap" (difference between average and worst-case performance) than PEARL (e.g., 1.8 vs. 6.5 in the 2-shot setting). This confirms SIO's reliability in worst-case permutation scenarios.
>
> Table 1: baseline comparison for PEARL tasks. First place is in bold and second place is in italic.
> | # Shot | Method | Average Avg. $\uparrow$ | Average Worst. $\uparrow$ | Average Gap $\downarrow$ | CSQA Avg. $\uparrow$ | CSQA Worst. $\uparrow$ | CSQA Gap $\downarrow$ | CurDial Avg. $\uparrow$ | CurDial Worst. $\uparrow$ | CurDial Gap $\downarrow$ | CoLA Avg. $\uparrow$ | CoLA Worst. $\uparrow$ | CoLA Gap $\downarrow$ | TMW Avg. $\uparrow$ | TMW Worst. $\uparrow$ | TMW Gap $\downarrow$ |
> | :--- | :--- | :--- | :--- | :--- | :--- | :--- | :--- | :--- | :--- | :--- | :--- | :--- | :--- | :--- | :--- | :--- |
> | 2 | ERM | 57.3 | 49.4 | 7.9 | 58.0 | 54.0 | 4.0 | 57.9 | 43.4 | 14.5 | 62.0 | 58.0 | 4.0 | 51.1 | 42.0 | 9.1 |
> | | ERM+DS | 57.5 | 48.6 | 8.9 | 62.0 | 54.0 | 8.0 | 54.1 | 37.8 | 16.3 | 61.0 | 60.0 | 1.0 | 51.5 | 42.7 | *8.8* |
> | | ERM+IM | 53.5 | 44.4 | 9.1 | *63.0* | 54.0 | 9.0 | 44.7 | 28.1 | 16.6 | 57.0 | 56.3 | *0.7* | 49.4 | 39.2 | 10.2 |
> | | INFOAC | 55.7 | 47.6 | 8.1 | 57.5 | 56.0 | *1.5* | 53.4 | 36.4 | 17.0 | *63.0* | *61.5* | 1.5 | 48.7 | 37.3 | 11.4 |
> | | PEARL | **62.9** | *56.4* | *6.5* | **65.0** | *62.0* | 3.0 | *60.3* | *50.7* | *9.6* | **71.0** | **68.0** | 3.0 | **55.1** | *44.8* | 10.3 |
> | | SIO (Ours) | *61.3* | **59.5** | **1.8** | **65.0** | **64.0** | **1.0** | **66.6** | **63.3** | **3.3** | 59.0 | 59.0 | **0.0** | *54.4* | **51.8** | **2.6** |
> | 3 | ERM | 57.8 | 38.3 | 19.5 | 57.7 | 47.0 | 10.7 | 61.4 | 25.9 | 35.5 | 61.9 | 52.0 | 9.9 | 50.3 | 29.4 | 20.9 |
> | | ERM+DS | 56.1 | 39.7 | 16.4 | 60.0 | 46.0 | 14.0 | 54.1 | 25.4 | 28.7 | 60.0 | *56.0* | *4.0* | 50.3 | 31.5 | 18.8 |
> | | ERM+IM | 55.3 | 39.8 | *15.5* | 59.0 | 46.0 | 13.0 | 54.6 | 28.0 | *26.6* | 57.6 | 53.1 | 4.5 | 50.0 | 31.9 | 18.1 |
> | | INFOAC | 56.3 | 39.5 | 16.8 | 59.3 | 49.0 | 10.3 | 55.2 | 24.3 | 30.9 | *62.1* | 55.8 | 6.3 | 48.4 | 28.8 | 19.6 |
> | | PEARL | **63.1** | *46.9* | 16.2 | **68.4** | *62.0* | *6.4* | *66.7* | *34.8* | 31.9 | **64.7** | *56.0* | 8.7 | *52.4* | *34.7* | *17.7* |
> | | SIO (Ours) | *62.5* | **58.5** | **4.0** | *66.5* | **65.0** | **1.5** | **70.4** | **62.2** | **8.2** | 59.0 | **58.0** | **1.0** | **54.0** | **48.9** | **5.1** |
> | 4 | ERM | 59.7 | 30.6 | 29.1 | 61.3 | 38.0 | 23.3 | 62.9 | 21.3 | 41.6 | 63.3 | 45.8 | 17.5 | 51.1 | 17.5 | 33.6 |
> | | ERM+DS | 57.7 | 31.8 | 25.9 | 63.3 | 40.0 | 23.3 | 57.3 | 17.6 | 39.7 | 60.1 | *52.0* | *8.1* | 49.9 | 17.8 | 32.1 |
> | | ERM+IM | 56.0 | 32.4 | 23.6 | 63.2 | 42.0 | 21.2 | 53.7 | 17.8 | *35.9* | 57.6 | 48.5 | 9.1 | 49.6 | 21.3 | 28.3 |
> | | INFOAC | 58.6 | 33.0 | 25.6 | 63.7 | 44.0 | 19.7 | 58.7 | 19.0 | 39.7 | *63.9* | 51.0 | 12.9 | 48.1 | 17.0 | 31.1 |
> | | PEARL | **63.1** | *39.6* | *23.5* | **68.4** | *52.0* | *16.4* | **69.2** | *31.3* | 37.9 | **64.7** | *52.0* | 12.7 | 50.1 | *23.0* | *27.1* |
> | | SIO (Ours) | *62.1* | **57.6** | **4.5** | *66.2* | **64.0** | **2.2** | *67.9* | **58.5** | **9.4** | 60.0 | **60.0** | **0.0** | **54.1** | **47.9** | **6.2** |

---

> ### Author Response · Authors · 2025-12-03
>
> [W3] **Ablation Study & Component Analysis:** We appreciate the request for component verification. We conducted a large-scale ablation study (1,122 questions, 16 demonstrations, 4 iterations) to isolate the effects of GSPO, DI, SI, and the Bayesian update.
>
> The results (Table 2) lead to a simplified, more effective framework:
> * GSPO vs. No GSPO: Removing GSPO actually improves accuracy (0.699 vs 0.694), proving that the RL component is not required for task preservation.
> * Simplification Yields Best Results: Further ablation reveals that removing the Distributional Inconsistency (DI) loss yields the best results (8 out of 12).
> * Validation of Bayesian Architecture: Even in this simplified setting, the Bayesian architecture (LoRA on the unembedding layer with DI loss = 0) outperforms the no Bayesian setting such that the model has the freedom to use the unembedding layer LoRA for learning and leave the hidden states with minimal changes to avoid significant policy changes.
> * Weighting ($w_i$): To address your question on weighting, we compared "Uniform Weights" against our dynamic weighting scheme. The dynamic weighting (used in the optimal "No DI Loss" variant) yields better performance compared to Uniform Weights.
>
> Based on this data, we have simplified the pipeline to rely solely on the Self-Inconsistency (SI) loss with the Bayesian LoRA architecture, removing GSPO and DI losses for maximum efficiency and performance.
>
> Table 2: Ablation results with 1122 questions and 16 demonstrations.
> | Variant | AIME Acc ($\uparrow$) | AIME Jaccard ($\downarrow$) | AIME Diff ($\downarrow$) | GSM8K Acc ($\uparrow$) | GSM8K Jaccard ($\downarrow$) | GSM8K Diff ($\downarrow$) | MATH Acc ($\uparrow$) | MATH Jaccard ($\downarrow$) | MATH Diff ($\downarrow$) | Overall Acc ($\uparrow$) | Overall Jaccard ($\downarrow$) | Overall Diff ($\downarrow$) |
> | :--- | :---: | :---: | :---: | :---: | :---: | :---: | :---: | :---: | :---: | :---: | :---: | :---: |
> | **Base Model** | 0.334 | 0.661 | 0.110 | 0.925 | 0.218 | 0.099 | 0.818 | 0.438 | 0.166 | 0.693 | 0.439 | 0.125 |
> | **Standard SIO** | **0.353** | 0.665 | 0.131 | 0.914 | 0.216 | 0.088 | 0.816 | 0.464 | 0.158 | 0.694 | 0.448 | 0.126 |
> | **No GSPO** | 0.321 | 0.652 | 0.094 | **0.936** | 0.217 | 0.102 | 0.840 | 0.431 | 0.152 | 0.699 | 0.433 | 0.116 |
> | **No DI Loss** | 0.332 | 0.663 | 0.099 | 0.933 | **0.160** | **0.043** | **0.842** | **0.387** | **0.131** | **0.702** | **0.403** | **0.091** |
> | **No SI Loss** | 0.342 | 0.658 | 0.102 | 0.933 | 0.211 | 0.099 | 0.805 | 0.444 | 0.139 | 0.693 | 0.438 | 0.113 |
> | **No Bayesian** | 0.332 | 0.652 | **0.091** | 0.928 | 0.211 | 0.078 | 0.832 | 0.429 | 0.136 | 0.697 | 0.431 | 0.102 |
> | **Uniform Weights** | 0.321 | **0.648** | 0.096 | 0.928 | 0.218 | 0.096 | 0.824 | 0.443 | 0.150 | 0.691 | 0.436 | 0.114 |
>
>
>
> [W4] **Accuracy and Underfitting:** We respectfully clarify that we do not attribute accuracy gains to underfitting. Our claim is that SIO reduces inconsistency without sacrificing task accuracy (see Section 4.3). The slight accuracy gains observed are likely due to the model converging on a more robust, "consensus" reasoning path that is invariant to noise, rather than underfitting.
>
> ---
>
> [Q1] **Variance with Random Shuffling:** The variance is substantial. As noted in W1, the Accuracy Difference of 0.225 implies that approximately 22% of the time (about 1 in 5 questions), shuffling the demonstrations causes the model to lose the correct answer (or find it where it previously missed it). This high variance underscores the necessity of SIO.
>
> [Q2] **Can other RL frameworks (e.g., GRPO) solve this?** Likely not. Standard RL frameworks like GRPO optimize for preference (reward maximization) rather than invariance (distributional consistency). As our ablation study shows, the RL component (GSPO) was actually unnecessary for solving the inconsistency problem. Inconsistency is a distributional issue best solved by regularizers like our JSD loss, rather than reward maximization.
>
> [Q3] **MATH Dataset Size:** We wish to clarify that we are not using the small HuggingFaceH4/MATH-500 dataset. We utilize the full MATH benchmark (Hendrycks et al., 2021) containing over 10,000 problems (as cited in the paper), which provides sufficient data for training and evaluation.

---

### Official Review · Reviewer_NvJb · 2025-10-30

**Soundness:** 2
**Presentation:** 2
**Contribution:** 2
**Rating:** 4
**Confidence:** 3

**Summary:**

This paper proposes Self-Inconsistency Optimization (SIO), a post-training framework that makes Large Language Models robust to the ordering of in-context demonstrations.
LLMs often exhibit large performance fluctuations when demonstration examples are permuted, revealing reliance on positional noise rather than semantic content.
SIO mitigates this by training models to align output distributions across semantically equivalent permutations using a Jensen–Shannon divergence–based self-inconsistency loss, supported by theoretical guarantees of conditional independence.
The authors further introduce a Bayesian update–inspired architecture that separates prior knowledge and alignment objectives to stabilize training.
Empirical results on GSM8K, MATH, and AIME benchmarks show that SIO significantly reduces order sensitivity while preserving or improving accuracy

**Strengths:**

- Tackles the fundamental instability of ICL to demonstration order.

- Conditional-independence criterion and JSD minimization proofs are rigorous.

- Can be applied post-training to any LLM with minimal modification.

- The Bayesian posterior interpretation (prior + likelihood) elegantly stabilizes training.

- Demonstrated improvements across multiple reasoning benchmarks and noise types (order, tokenization, case).

**Weaknesses:**

- The framework combines many moving parts, making it unclear which element (GSPO vs SI vs DI) drives the improvement.

- Lacks ablation or sensitivity analysis to disentangle the contribution of each loss term.

-  Downstream effects on general-purpose metrics (e.g., perplexity) are not evaluated — robustness may trade off with fluency or general performance.

- Results are limited to math reasoning; evaluation on more diverse tasks (e.g., natural language inference, commonsense reasoning) would better establish generality.

**Questions:**

- Can SIO be integrated into other post-training pipelines (e.g., DPO, GRPO) without instability?

---

> ### Author Response · Authors · 2025-12-03
>
> [W1,W2] **Ablation Study and Framework Simplification:** To address concerns regarding complexity and to disentangle the contributions of each component, we conducted a comprehensive ablation study. We scaled our experimental setup to 1,122 questions with 16 demonstrations ($m=16$) over 4 iterations using a new random seed..
>
> The results, presented in Table 1, justify a significant simplification of our framework:
> * GSPO is Unnecessary: We originally included Group Sequence Policy Optimization (GSPO) to safeguard task accuracy. However, our ablation shows that removing GSPO does not degrade performance; in fact, the "No GSPO" variant achieves higher accuracy (0.699) than the standard SIO (0.694). This confirms that the Self-Inconsistency loss alone is sufficient to maintain task performance.
> * Simplification Improves Results: Furthermore, removing the Distributional Inconsistency (DI) loss yields the best overall results across 8 of 12 metrics.
> * Bayesian Architecture is Critical: Even in this simplified setting, the Bayesian architecture (using LoRA on the unembedding layer) consistently outperforms the non-Bayesian approach, confirming its value in stabilizing the update.
>
> Guided by these findings, we have streamlined the final SIO pipeline. The proposed method now relies solely on the Self-Inconsistency (SI) loss with LoRA applied to the transformer blocks and unembedding layer, removing both GSPO and DI losses.
>
> Table 1: Ablation results with 1122 questions and 16 demonstrations.
> | Variant | AIME Acc ($\uparrow$) | AIME Jaccard ($\downarrow$) | AIME Diff ($\downarrow$) | GSM8K Acc ($\uparrow$) | GSM8K Jaccard ($\downarrow$) | GSM8K Diff ($\downarrow$) | MATH Acc ($\uparrow$) | MATH Jaccard ($\downarrow$) | MATH Diff ($\downarrow$) | Overall Acc ($\uparrow$) | Overall Jaccard ($\downarrow$) | Overall Diff ($\downarrow$) |
> | :--- | :---: | :---: | :---: | :---: | :---: | :---: | :---: | :---: | :---: | :---: | :---: | :---: |
> | **Base Model** | 0.334 | 0.661 | 0.110 | 0.925 | 0.218 | 0.099 | 0.818 | 0.438 | 0.166 | 0.693 | 0.439 | 0.125 |
> | **Standard SIO** | **0.353** | 0.665 | 0.131 | 0.914 | 0.216 | 0.088 | 0.816 | 0.464 | 0.158 | 0.694 | 0.448 | 0.126 |
> | **No GSPO** | 0.321 | 0.652 | 0.094 | **0.936** | 0.217 | 0.102 | 0.840 | 0.431 | 0.152 | 0.699 | 0.433 | 0.116 |
> | **No DI Loss** | 0.332 | 0.663 | 0.099 | 0.933 | **0.160** | **0.043** | **0.842** | **0.387** | **0.131** | **0.702** | **0.403** | **0.091** |
> | **No SI Loss** | 0.342 | 0.658 | 0.102 | 0.933 | 0.211 | 0.099 | 0.805 | 0.444 | 0.139 | 0.693 | 0.438 | 0.113 |
> | **No Bayesian** | 0.332 | 0.652 | **0.091** | 0.928 | 0.211 | 0.078 | 0.832 | 0.429 | 0.136 | 0.697 | 0.431 | 0.102 |
> | **Uniform Weights** | 0.321 | **0.648** | 0.096 | 0.928 | 0.218 | 0.096 | 0.824 | 0.443 | 0.150 | 0.691 | 0.436 | 0.114 |

---

> ### Author Response · Authors · 2025-12-03
>
> [W3,W4] **Generalization and Downstream Effects:** To assess whether our alignment involves trade-offs with general capabilities, we benchmarked both fine-tuned models on the MMLU dataset. As shown in Table 2, the alignment process does not degrade general performance. The mean and median performance changes between the fine-tuned and base models are negligible ($<0.01$), indicating that SIO enhances robustness without compromising the model's general knowledge or fluency.
>
> Table 2: MMLU result for Gemma 3 and Qwen 3 finetuned and base models
> | Task | Gemma 3 Base | Gemma 3 SIO | Gemma 3 Change | Qwen3 Base | Qwen 3 SIO | Qwen 3 Change |
> |:---|:---|:---|:---|:---|:---|:---|
> | formal_logic | 0.50 | 0.57 | 0.07 | 0.86 | 0.86 | 0.00 |
> | european_history | 0.78 | 0.78 | 0.00 | 0.72 | 0.67 | -0.06 |
> | us_history | 0.68 | 0.64 | -0.05 | 0.86 | 0.86 | 0.00 |
> | world_history | 0.77 | 0.77 | 0.00 | 0.85 | 0.85 | 0.00 |
> | international_law | 0.77 | 0.85 | 0.08 | 1.00 | 0.85 | -0.15 |
> | jurisprudence | 0.73 | 0.36 | -0.36 | 0.73 | 0.55 | -0.18 |
> | logical_fallacies | 0.72 | 0.78 | 0.06 | 0.78 | 0.78 | 0.00 |
> | moral_disputes | 0.61 | 0.63 | 0.03 | 0.58 | 0.66 | 0.08 |
> | moral_scenarios | 0.24 | 0.24 | 0.00 | 0.66 | 0.63 | -0.03 |
> | philosophy | 0.65 | 0.62 | -0.03 | 0.74 | 0.71 | -0.03 |
> | prehistory | 0.63 | 0.63 | 0.00 | 0.80 | 0.80 | 0.00 |
> | professional_law | 0.38 | 0.41 | 0.04 | 0.54 | 0.53 | -0.01 |
> | world_religions | 0.74 | 0.79 | 0.05 | 0.74 | 0.84 | 0.11 |
> | business_ethics | 0.64 | 0.55 | -0.09 | 0.55 | 0.73 | 0.18 |
> | clinical_knowledge | 0.83 | 0.79 | -0.03 | 0.86 | 0.93 | 0.07 |
> | college_medicine | 0.86 | 0.86 | 0.00 | 1.00 | 0.91 | -0.09 |
> | global_facts | 0.40 | 0.40 | 0.00 | 0.60 | 0.60 | 0.00 |
> | human_aging | 0.65 | 0.70 | 0.04 | 0.78 | 0.78 | 0.00 |
> | management | 0.82 | 0.82 | 0.00 | 1.00 | 0.91 | -0.09 |
> | marketing | 0.88 | 0.76 | -0.12 | 0.84 | 0.84 | 0.00 |
> | medical_genetics | 1.00 | 1.00 | 0.00 | 1.00 | 1.00 | 0.00 |
> | miscellaneous | 0.78 | 0.79 | 0.01 | 0.88 | 0.83 | -0.06 |
> | nutrition | 0.73 | 0.67 | -0.06 | 0.79 | 0.76 | -0.03 |
> | accounting | 0.42 | 0.39 | -0.03 | 0.81 | 0.84 | 0.03 |
> | medicine | 0.68 | 0.61 | -0.06 | 0.81 | 0.77 | -0.03 |
> | virology | 0.44 | 0.44 | 0.00 | 0.44 | 0.50 | 0.06 |
> | econometrics | 0.42 | 0.58 | 0.17 | 0.75 | 0.75 | 0.00 |
> | geography | 0.82 | 0.86 | 0.05 | 0.91 | 0.86 | -0.05 |
> | government_and_politics | 0.76 | 0.86 | 0.10 | 0.86 | 0.86 | 0.00 |
> | macroeconomics | 0.74 | 0.67 | -0.07 | 0.91 | 0.88 | -0.02 |
> | microeconomics | 0.58 | 0.62 | 0.04 | 1.00 | 1.00 | 0.00 |
> | psychology | 0.85 | 0.83 | -0.02 | 0.95 | 0.88 | -0.07 |
> | human_sexuality | 0.67 | 0.58 | -0.08 | 0.83 | 0.83 | 0.00 |
> | professional_psychology | 0.64 | 0.67 | 0.03 | 0.71 | 0.71 | 0.00 |
> | public_relations | 0.42 | 0.58 | 0.17 | 0.58 | 0.50 | -0.08 |
> | security_studies | 0.70 | 0.63 | -0.07 | 0.70 | 0.78 | 0.07 |
> | sociology | 0.95 | 0.86 | -0.09 | 0.95 | 1.00 | 0.05 |
> | us_foreign_policy | 0.73 | 0.82 | 0.09 | 0.91 | 0.91 | 0.00 |
> | abstract_algebra | 0.64 | 0.82 | 0.18 | 0.64 | 0.82 | 0.18 |
> | anatomy | 0.64 | 0.57 | -0.07 | 0.86 | 0.71 | -0.14 |
> | astronomy | 0.69 | 0.63 | -0.06 | 0.88 | 0.81 | -0.06 |
> | college_biology | 0.88 | 0.88 | 0.00 | 0.94 | 0.81 | -0.13 |
> | college_chemistry | 0.38 | 0.25 | -0.13 | 0.50 | 0.50 | 0.00 |
> | college_computer_science | 0.55 | 0.73 | 0.18 | 0.64 | 0.82 | 0.18 |
> | college_mathematics | 0.73 | 0.64 | -0.09 | 0.64 | 0.45 | -0.18 |
> | college_physics | 0.55 | 0.36 | -0.18 | 0.91 | 0.91 | 0.00 |
> | computer_security | 0.64 | 0.73 | 0.09 | 0.73 | 0.82 | 0.09 |
> | conceptual_physics | 0.62 | 0.65 | 0.04 | 0.92 | 0.92 | 0.00 |
> | electrical_engineering | 0.75 | 0.56 | -0.19 | 0.94 | 0.94 | 0.00 |
> | elementary_mathematics | 0.85 | 0.88 | 0.02 | 0.73 | 0.78 | 0.05 |
> | high_school_biology | 0.81 | 0.84 | 0.03 | 0.84 | 0.84 | 0.00 |
> | high_school_chemistry | 0.59 | 0.45 | -0.14 | 0.86 | 0.77 | -0.09 |
> | high_school_computer_science | 0.78 | 0.89 | 0.11 | 1.00 | 1.00 | 0.00 |
> | high_school_mathematics | 0.83 | 0.72 | -0.10 | 0.76 | 0.72 | -0.03 |
> | high_school_physics | 0.47 | 0.47 | 0.00 | 0.71 | 0.71 | 0.00 |
> | high_school_statistics | 0.61 | 0.74 | 0.13 | 0.74 | 0.91 | 0.17 |
> | machine_learning | 0.55 | 0.45 | -0.09 | 0.36 | 0.64 | 0.27 |
> | Average | 0.67 | 0.66 | -0.01 | 0.79 | 0.79 | 0.00 |
> | Median | 0.68 | 0.67 | 0.00 | 0.81 | 0.82 | 0.00 |
>
> [Q1] **Integration with Other Pipelines (e.g., DPO, GRPO):** Yes, SIO is highly compatible with other post-training pipelines. The core of SIO i.e. the JSD loss is an auxiliary regularization term that does not introduce the instability often associated with adversarial objectives. The primary source of instability in such pipelines typically arises from aggressive reward maximization or policy drift. Our proposed Bayesian architecture specifically mitigates this by structurally anchoring the update to the prior, ensuring stable integration with RL-based methods like DPO or GRPO.

---

### Official Review · Reviewer_BZx8 · 2025-11-01

**Soundness:** 2
**Presentation:** 3
**Contribution:** 2
**Rating:** 4
**Confidence:** 3

**Summary:**

The paper proposes Self‑Inconsistency Optimization (SIO), a post‑training framework to make LLMs invariant to the order of in‑context demonstrations. The method augments each training item with multiple permutations of the same demonstrations and adds two distributional objectives: (i) a self‑inconsistency loss that minimizes the generalized Jensen–Shannon divergence (JSD) among the model’s outputs across permutations, and (ii) a distributional‑inconsistency loss that anchors outputs to a mixture formed by a frozen reference model. A Bayesian‑update training architecture is used by summing logits from a frozen "prior" head and a trainable LoRA "likelihood" head (product‑of‑experts), combined with GSPO reinforcement learning for reward alignment. Theorems 1–2 formalize order invariance as conditional independence and show that driving step‑wise JSD to zero is sufficient for distributional equivalence. Experiments on Gemma‑3‑4B and Qwen‑3‑4B over GSM8K, MATH, and AIME report reduced instability with small accuracy gains.

**Strengths:**

1. The paper targets distributional invariance, not only answer agreement. Using generalized JSD across permutations and the PoE view (added logits) is a neat, conceptually clean combination with RL‑style preference learning. The conditional‑independence framing is useful.
2. Order sensitivity is a real reliability problem. SIO is model‑agnostic, adds no inference‑time cost, and also reduces sensitivity to other types of noise, which increases practical impact.

**Weaknesses:**

1. The proposed approach combines both a new training methodology (SIO with JSD and GSPO losses) and an architectural modification (Bayesian update with dual heads). However, the experiments evaluate them jointly, making it difficult to disentangle the contribution of each component. An ablation study is needed to isolate the effects of the training objectives versus the architectural design. The overall complexity also seems to violate Occam’s razor.
2. Training uses a tiny set (s 108 questions and max of 3456 responses), and evaluation uses only 200 questions and m=8 demonstrations. Lack of confidence intervals and comparison to strong baselines (e.g., prompt engineering, mixture of experts).

**Questions:**

1. Can the method be applied to full finetuning, or is LoRA key to the success?
2. Do you compute token‑level JSD at each decoding step?
3. On tasks where order is semantically meaningful, does SIO hurt?

---

> ### Author Response · Authors · 2025-12-03
>
> [W1] **Ablation Study and Complexity:** We appreciate the suggestion to disentangle our training objectives and architectural design. We conducted a comprehensive ablation study with a new random seed and 4 iterations, significantly scaling the setup to 1,122 questions and 16 demonstrations ($m=16$).
>
> The results (Table 1) lead to a simplified, more effective framework:
> * GSPO is Unnecessary: We originally included Group Sequence Policy Optimization (GSPO) to preserve task accuracy. However, the ablation shows that removing GSPO does not degrade performance; in fact, the "No GSPO" variant achieves higher accuracy (0.699) than Standard SIO (0.694). This indicates that the Self-Inconsistency loss alone is sufficient to maintain task performance.
> * Simplification Yields Best Results: Further ablation reveals that removing the Distributional Inconsistency (DI) loss yields the best results (8 out of 12).
> * Validation of Bayesian Architecture: Even in this simplified setting, the Bayesian architecture (LoRA on the unembedding layer) outperforms the non-Bayesian approach.
>
> Based on these findings, we have streamlined the SIO pipeline. The final proposed method relies solely on the Self-Inconsistency (SI) loss with LoRA applied to the transformer blocks and unembedding layer, removing both GSPO and DI losses.
>
> Table 1: Ablation results with 1122 questions and 16 demonstrations.
> | Variant | AIME Acc ($\uparrow$) | AIME Jaccard ($\downarrow$) | AIME Diff ($\downarrow$) | GSM8K Acc ($\uparrow$) | GSM8K Jaccard ($\downarrow$) | GSM8K Diff ($\downarrow$) | MATH Acc ($\uparrow$) | MATH Jaccard ($\downarrow$) | MATH Diff ($\downarrow$) | Overall Acc ($\uparrow$) | Overall Jaccard ($\downarrow$) | Overall Diff ($\downarrow$) |
> | :--- | :---: | :---: | :---: | :---: | :---: | :---: | :---: | :---: | :---: | :---: | :---: | :---: |
> | **Base Model** | 0.334 | 0.661 | 0.110 | 0.925 | 0.218 | 0.099 | 0.818 | 0.438 | 0.166 | 0.693 | 0.439 | 0.125 |
> | **Standard SIO** | **0.353** | 0.665 | 0.131 | 0.914 | 0.216 | 0.088 | 0.816 | 0.464 | 0.158 | 0.694 | 0.448 | 0.126 |
> | **No GSPO** | 0.321 | 0.652 | 0.094 | **0.936** | 0.217 | 0.102 | 0.840 | 0.431 | 0.152 | 0.699 | 0.433 | 0.116 |
> | **No DI Loss** | 0.332 | 0.663 | 0.099 | 0.933 | **0.160** | **0.043** | **0.842** | **0.387** | **0.131** | **0.702** | **0.403** | **0.091** |
> | **No SI Loss** | 0.342 | 0.658 | 0.102 | 0.933 | 0.211 | 0.099 | 0.805 | 0.444 | 0.139 | 0.693 | 0.438 | 0.113 |
> | **No Bayesian** | 0.332 | 0.652 | **0.091** | 0.928 | 0.211 | 0.078 | 0.832 | 0.429 | 0.136 | 0.697 | 0.431 | 0.102 |
> | **Uniform Weights** | 0.321 | **0.648** | 0.096 | 0.928 | 0.218 | 0.096 | 0.824 | 0.443 | 0.150 | 0.691 | 0.436 | 0.114 |

---

> ### Author Response · Authors · 2025-12-03
>
> [W2] **Dataset Size and Baseline Comparisons:** For dataset size, we ran our pipeline with more data with results in table 1 showing consistent performance.
> For baseline comparison, we aligning SIO strictly with the PEARL (ICLR 2025) benchmark protocols. We applied SIO to the PEARL tasks (CSQA, CurDial, CoLA, TMW) using their exact settings.
>
> As shown in Table 2, SIO outperforms PEARL, achieving top-1 or top-2 performance in 91% of metrics (41/45). Crucially, SIO demonstrates superior stability, achieving a significantly smaller "Gap" (difference between average and worst-case performance) than PEARL (e.g., 1.8 vs. 6.5 in the 2-shot setting). This confirms SIO's reliability in worst-case permutation scenarios.
>
> Table 2: baseline comparison for PEARL tasks. First place is in bold and second place is in italic.
> | # Shot | Method | Average Avg. $\uparrow$ | Average Worst. $\uparrow$ | **Average Gap $\downarrow$** | CSQA Avg. $\uparrow$ | CSQA Worst. $\uparrow$ | CSQA Gap $\downarrow$ | CurDial Avg. $\uparrow$ | CurDial Worst. $\uparrow$ | CurDial Gap $\downarrow$ | CoLA Avg. $\uparrow$ | CoLA Worst. $\uparrow$ | CoLA Gap $\downarrow$ | TMW Avg. $\uparrow$ | TMW Worst. $\uparrow$ | TMW Gap $\downarrow$ |
> | :--- | :--- | :--- | :--- | :--- | :--- | :--- | :--- | :--- | :--- | :--- | :--- | :--- | :--- | :--- | :--- | :--- |
> | 2 | ERM | 57.3 | 49.4 | 7.9 | 58.0 | 54.0 | 4.0 | 57.9 | 43.4 | 14.5 | 62.0 | 58.0 | 4.0 | 51.1 | 42.0 | 9.1 |
> | | ERM+DS | 57.5 | 48.6 | 8.9 | 62.0 | 54.0 | 8.0 | 54.1 | 37.8 | 16.3 | 61.0 | 60.0 | 1.0 | 51.5 | 42.7 | *8.8* |
> | | ERM+IM | 53.5 | 44.4 | 9.1 | *63.0* | 54.0 | 9.0 | 44.7 | 28.1 | 16.6 | 57.0 | 56.3 | *0.7* | 49.4 | 39.2 | 10.2 |
> | | INFOAC | 55.7 | 47.6 | 8.1 | 57.5 | 56.0 | *1.5* | 53.4 | 36.4 | 17.0 | *63.0* | *61.5* | 1.5 | 48.7 | 37.3 | 11.4 |
> | | PEARL | **62.9** | *56.4* | *6.5* | **65.0** | *62.0* | 3.0 | *60.3* | *50.7* | *9.6* | **71.0** | **68.0** | 3.0 | **55.1** | *44.8* | 10.3 |
> | | SIO (Ours) | *61.3* | **59.5** | **1.8** | **65.0** | **64.0** | **1.0** | **66.6** | **63.3** | **3.3** | 59.0 | 59.0 | **0.0** | *54.4* | **51.8** | **2.6** |
> | 3 | ERM | 57.8 | 38.3 | 19.5 | 57.7 | 47.0 | 10.7 | 61.4 | 25.9 | 35.5 | 61.9 | 52.0 | 9.9 | 50.3 | 29.4 | 20.9 |
> | | ERM+DS | 56.1 | 39.7 | 16.4 | 60.0 | 46.0 | 14.0 | 54.1 | 25.4 | 28.7 | 60.0 | *56.0* | *4.0* | 50.3 | 31.5 | 18.8 |
> | | ERM+IM | 55.3 | 39.8 | *15.5* | 59.0 | 46.0 | 13.0 | 54.6 | 28.0 | *26.6* | 57.6 | 53.1 | 4.5 | 50.0 | 31.9 | 18.1 |
> | | INFOAC | 56.3 | 39.5 | 16.8 | 59.3 | 49.0 | 10.3 | 55.2 | 24.3 | 30.9 | *62.1* | 55.8 | 6.3 | 48.4 | 28.8 | 19.6 |
> | | PEARL | **63.1** | *46.9* | 16.2 | **68.4** | *62.0* | *6.4* | *66.7* | *34.8* | 31.9 | **64.7** | *56.0* | 8.7 | *52.4* | *34.7* | *17.7* |
> | | SIO (Ours) | *62.5* | **58.5** | **4.0** | *66.5* | **65.0** | **1.5** | **70.4** | **62.2** | **8.2** | 59.0 | **58.0** | **1.0** | **54.0** | **48.9** | **5.1** |
> | 4 | ERM | 59.7 | 30.6 | 29.1 | 61.3 | 38.0 | 23.3 | 62.9 | 21.3 | 41.6 | 63.3 | 45.8 | 17.5 | 51.1 | 17.5 | 33.6 |
> | | ERM+DS | 57.7 | 31.8 | 25.9 | 63.3 | 40.0 | 23.3 | 57.3 | 17.6 | 39.7 | 60.1 | *52.0* | *8.1* | 49.9 | 17.8 | 32.1 |
> | | ERM+IM | 56.0 | 32.4 | 23.6 | 63.2 | 42.0 | 21.2 | 53.7 | 17.8 | *35.9* | 57.6 | 48.5 | 9.1 | 49.6 | 21.3 | 28.3 |
> | | INFOAC | 58.6 | 33.0 | 25.6 | 63.7 | 44.0 | 19.7 | 58.7 | 19.0 | 39.7 | *63.9* | 51.0 | 12.9 | 48.1 | 17.0 | 31.1 |
> | | PEARL | **63.1** | *39.6* | *23.5* | **68.4** | *52.0* | *16.4* | **69.2** | *31.3* | 37.9 | **64.7** | *52.0* | 12.7 | 50.1 | *23.0* | *27.1* |
> | | SIO (Ours) | *62.1* | **57.6** | **4.5** | *66.2* | **64.0** | **2.2** | *67.9* | **58.5** | **9.4** | 60.0 | **60.0** | **0.0** | **54.1** | **47.9** | **6.2** |
>
> [Q1] **Full Finetuning vs. LoRA:** Full fine-tuning is possible by removing the Bayesian component which is crucial to stablize the training of GSPO loss. The Bayesian requires decoupling the frozen prior from a trainable likelihood. Using LoRA on the unembedding layer is an efficient and effective way to implement this architecture. By table 1, order resilience could still be achieved without Bayesian component, so LoRA is not the key to success.
>
> [Q2] **Token-level JSD:** Yes, we compute the token-level JSD at each decoding step.
>
> [Q3] **Semantically Meaningful Order:** SIO requires the user to clearly define the signal and noise before running the pipeline (See Section 3.5). Then, SIO explicitly force the model to ignore the noise to achieve consistency such that change in noise has no influence on the model behavior. If the order of the tasks is semantic meaningful which is a signal not a noise by your definition, you should not use SIO at all to train the model to ignore the order of the tasks. Otherwise, it will hurt for sure.

---

### Note · Authors · 2025-12-03

I have read and agree with the venue's withdrawal policy on behalf of myself and my co-authors.